# Knowledge-based Visual Question Answer with Multimodal Processing, Retrieval and Filtering

**Yuyang Hong**[1,2]*, **Jiaqi Gu**[3]*, **Qi Yang**[1,2], **Lubin Fan**[3]†,

**Yue Wu**[3], **Ying Wang**[2], **Kun Ding**[2]†, **Shiming Xiang**[1,2], **Jieping Ye**[3]

[1]School of Artificial Intelligence, University of Chinese Academy of Sciences
[2]MAIS, Institute of Automation, Chinese Academy of Sciences
[3]Alibaba Cloud Computing

## Abstract

Knowledge-based visual question answering (KB-VQA) requires visual language models (VLMs) to integrate visual understanding with external knowledge retrieval. Although retrieval-augmented generation (RAG) achieves significant advances in this task by combining knowledge-base querying, it still struggles with the quality of multimodal queries and the relevance of retrieved results. To overcome these challenges, we propose a novel three-stage method, termed Wiki-PRF, including **P**rocessing, **R**etrieval and **F**iltering stages. The processing stage dynamically invokes visual tools to extract precise multimodal information for retrieval. The retrieval stage integrates visual and text features to achieve multimodal knowledge retrieval. The filtering stage performs relevance filtering and concentration on retrieval results. To this end, we introduce a visual language model trained with answer accuracy and format consistency as reward signals via a reinforcement learning manner. This enhances the model's reasoning, tool invocation for accurate queries, and filtering of irrelevant content. Experiments on benchmark datasets (E-VQA and InfoSeek) show significant improvements (36.0 and 42.8) in answer quality, achieving state-of-the-art performance. Code is available at: `https://github.com/cqu-student/Wiki-PRF`

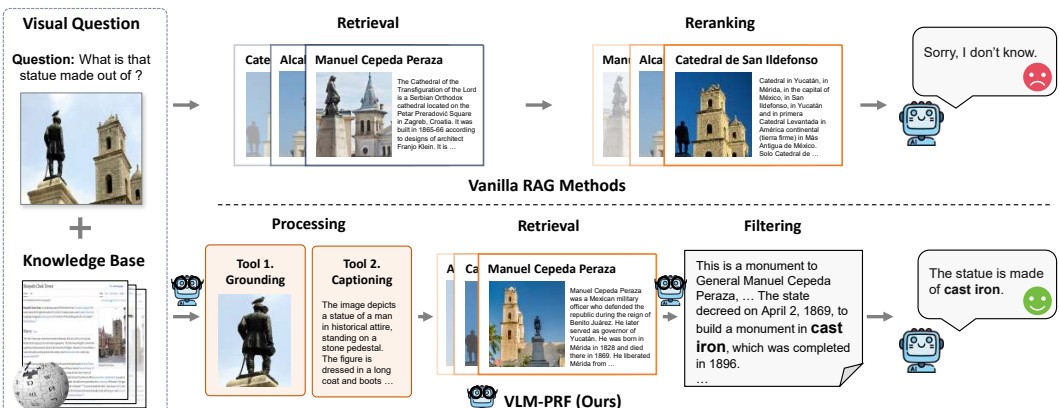

Figure 1: **Illustration of vanilla RAG methods and our Wiki-PRF.** Different from the traditional RAG methods (above), our method (below) employs multimodal tools processing stage and a further filtering stage, enabling more effective retrieval and extraction of task-relevant information.

---

[1]Equal contribution: Yuyang Hong <hongyuyang2023@ia.ac.cn>, Jiaqi Gu <vadin@zju.edu.cn>

[2]Corresponding authors: Lubin Fan <lubin.flb@alibaba-inc.com>, Kun Ding <kun.ding@ia.ac.cn>

39th Conference on Neural Information Processing Systems (NeurIPS 2025).

# 1 Introduction

Visual Language Models (VLMs) [1, 2, 3] have demonstrated remarkable capabilities in Visual Question Answering (VQA) tasks [4, 5]. Despite their effectiveness, they still face challenges in addressing knowledge-based visual question answering (KB-VQA), as such questions require not only an understanding of visual content but also the integration of external knowledge. For instance, answering the question "What is that statue made of?" in Figure 1 requires factual knowledge that goes beyond the visual content. To address this issue, retrieval-augmented generation (RAG) methods enhance model performance by incorporating mechanisms to access and integrate external information. These methods typically involve three steps: first, retrieving knowledge based on the given visual or textual content; second, reranking the retrieved information for relevance; and third, generating answers using the most pertinent content.

However, current methods [6, 7, 8, 9, 10, 11] often fail to retrieve the most relevant information when handling complex visual content and questions, leading to suboptimal answers. This issue primarily arises from two key challenges. (1) Fine-grained knowledge retrieval in complex visual scenes: When process information-rich images, existing retrieval methods that rely on full-image visual features are often insufficiently precise for effective knowledge retrieval. For example, when asking about statues near a bell tower in Figure 1, the statue may occupy only a small portion of the image. Consequently, the retrieval process can be heavily influenced by the more prominent bell tower, introducing excessive irrelevant information related to it. (2) Precise filtering of irrelevant information from large scale retrieved results: After retrieving contextual information, it is difficult to filter out irrelevant or low-quality content using paragraph reranking alone. The retrieved results typically contain significant amounts of extraneous information, which can affect the accuracy of the generated answers.

To address these challenges, we propose a novel multimodal RAG method consisting of three stages: processing, retrieval, and filtering. The core idea is to obtain more relevant knowledge to generate accurate answers. For coarse retrieval limitations, we innovatively explored a tool-based fine-grained retrieval mechanism. For irrelevant information, we innovatively employ a question-based filtering stage. Specifically, in the processing stage, the visual language model autonomously invokes image-processing tools based on the input image and question. These tools perform operations such as image captioning, visual grounding, and image flipping to extract detailed information related to the question from the image, thereby generating high-quality multimodal retrieval inputs. In the retrieval stage, multimodal retrieval is conducted using both visual features and text descriptions to retrieve relevant knowledge. In the filtering stage, the retrieved contextual information is filtered and condensed to remove redundancy, extract the most relevant knowledge, and provide to the answer generator for generating accurate responses. To this end, we introduce Wiki-PRF, a RAG method that not only supports basic multimodal question-answering functionality but also enhances reasoning based on the input image and question. Wiki-PRF can flexibly invoke visual tools and demonstrates stronger capabilities in filtering and condensing retrieval results.

To enable the visual language model to possess the aforementioned reasoning ability, we train a VLM-PRF model using reinforcement learning (RL). This is because training data collected for complex visual question-answering tasks often lacks the intermediate reasoning steps, which are necessary for effective supervised fine-tuning of VLMs. RL [12, 13, 14], as a paradigm for learning strategies to achieve specific goals, has been widely adopted in recent years to enhance the reasoning capabilities of VLMs [15, 16, 17, 18, 19, 20] for specialized tasks. RL can utilize a small amount of sample data, relying solely on answer accuracy as the reward signal, to train the model to generate high-quality retrieval content by accurately invoking task-specific tools. Additionally, it enables the model to selectively retain and condense the most relevant retrieval results for the query. Specifically, we employ the LoRA [21] to train only a small number of additional parameters, enabling our Wiki-PRF to enhance its RAG capabilities without compromising its core question-answering abilities. In summary, our main contributions are as follows:

- A knowledge-based visual question-answering method using a Processing-Retrieval-Filtering framework is proposed, named Wiki-PRF. It effectively leverages external tools for information retrieval and systematically filters the retrieved knowledge to support the generation of precise answers.

- We introduce VLM-PRF, a visual language model for multimodal RAG tasks, trained via reinforcement learning to enhance reasoning. To our knowledge, this represents the first application of reinforcement learning to multimodal retrieval-augmented generation, requiring minimal training data while enabling flexible tool use and robust processing.

- Comprehensive experiments demonstrate that Wiki-PRF achieves state-of-the-art performance on E-VQA (36.0) and InfoSeek (42.8). Additional analyses further validate our method's effectiveness.

## 2 Related Work

### 2.1 Knowledge-based Visual Question Answering

Knowledge-Based Visual Question Answering (Knowledge-Based VQA) [22] As a critical branch of Visual Question Answering (VQA), Knowledge-Based VQA demands models to integrate the understanding of visual content and question with external knowledge bases for reasoning and answering. Based on knowledge base modalities, Knowledge-Based VQA frameworks can be categorized into unimodal [23, 24, 25, 26, 6, 7] and multimodal [27, 28, 8, 9, 10, 11] paradigms. Unimodal methods [26, 6, 7] typically utilize text-only datasets such as Wiki-21M[29] and GS112K[30] as external knowledge sources. For unimodal methods, TRiG[7] facilitates knowledge passage retrieval and generative question answering by converting images into plain text, thus fully harnessing the power of large-scale knowledge bases and pre-trained language models.
For multimodal methods [8, 9, 10, 11, 31], external knowledge bases typically incorporate datasets such as Encyclopedic VQA (E-VQA)[27] and InfoSeek[28], which include both Wikipedia images and corresponding textual information. MuKEA[11] represents multimodal knowledge through explicit triplets to capture the implicit relationships between visual objects and factual answers. EchoSight[9] first retrieves candidate Wikipedia articles using visual information, then re-ranks them based on text-image query relevance to improve retrieval performance. Unlike previous methods, our method enhances the utilization of external knowledge bases by enabling the model to autonomously select and filter relevant information during the retrieval processing.

### 2.2 Reinforcement Learning for Visual Language Model

Reinforcement learning (RL) [12, 13, 14], a learning paradigm that improves model decision-making through interaction with an environment and feedback in the form of rewards, has recently been widely applied to vision-language models [32, 2, 33, 34, 35] (VLMs). Some works [15, 16, 17, 18, 19, 20] focus on enhancing the reasoning capabilities of Vision-Language Models (VLMs) through reinforcement learning. R1-OneVision[20] framework innovatively bridges vision and language by encoding visual data into formalized textual representations, enabling robust and precise reasoning grounded in linguistic semantics. VisualThinker-R1-Zero[16] achieves the first successful realization of an 'Aha Moment' in multimodal reasoning using a 2B-parameter VLM. Other works[36, 37, 38, 39, 40] focus on leveraging reinforcement learning to improve the performance of Vision-Language Models in specific areas like mathematical reasoning and visual perception. Visual-RFT [39] leverages VLMs to generate reasoning-enhanced responses and integrates task-specific verifiable rewards (e.g., IoU for detection) via policy optimization methods like GRPO [41], improving model performance. In contrast to above work, our study uniquely introduces RAG capabilities into VLMs via RL. As far as we are aware, this is the first exploration of RL-based method for RAG in VLM.

## 3 Method

### 3.1 Overview

Knowledge-based VQA requires answering question $Q$ that is highly relevant to a given reference image $I$, with the assistance of a knowledge base KB. In our setup, KB $\in \{(a_1, I_1), ..., (a_n, I_n)\}$ consists of a million-scale collection of entity articles $\{a_i\}$ along with their corresponding image set $\{I_i\}$. Our goal is to improve multimodal retrieval quality by flexibly invoking visual tools and enhancing relevance through filtering and enrichment.

As illustrated in Figure 2, the overall architecture of Wiki-PRF consists of three key components: an external knowledge base (KB), a model (VLM-PRF) trained via reinforcement learning, and a

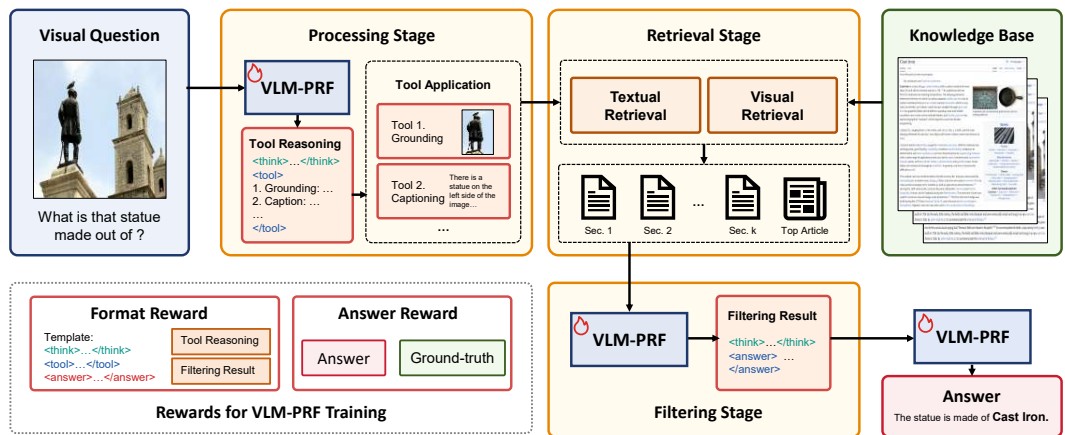

Figure 2: **Overview of Wiki-PRF**. Wiki-PRF comprises three key stages: (1) Processing Stage: VLM-PRF processes the input image and its corresponding question using external tools. (2) Retrieval Stage: Relevant Wikipedia articles are retrieved, split into individual sections, and ranked based on their similarity to the processed input. (3) Filtering Stage: The re-ranked article sections are further filtered by VLM-PRF to retain the most relevant content, which is then fed into the VLM for final answer generation. During training, VLM-PRF is supervised using two types of reward signals: answer reward , which evaluates the correctness of the generated answer, and format reward , which ensures the output adheres to the desired structure.

base model without extra trained parameters. The Wiki-PRF method comprises three main stages: (1) Processing Stage: The VLM-PRF model invokes external tools to process the raw reference image $I$ and question $Q$, generating precise retrieval queries $Query$. (2) Multimodal Retrieval Stage: The model performs multimodal information retrieval based on the generated query $Query$ and retrieves relevant information from the knowledge base. (3) Filtering Stage: The VLM-PRF model filters and extracts highly relevant information from the retrieval results and structures it into task-oriented knowledge, which is subsequently utilized to augment the answer.

## 3.2 Processing Stage

Previous methods [27, 28, 8, 9, 10, 11] rely on raw input for retrieval, often missing key details due to a lack of interactive processing. For instance, a statue next to a church might be overlooked in favor of the church itself. To address this, our method introduces tool-based preprocessing before retrieval, enhancing results through secondary data refinement. We employ several representative tools: 1) A captioning tool that captures high-level semantic information from images. 2) A grounding tool that extracts regions of interest for precise, detailed retrieval. 3) A flipping tool that adjusts the images orientation to mitigate the impact of direction on retrieval. Through these tools, Wiki-PRF achieves more comprehensive and accurate retrieval results. In essence, VLM-PRF provides the strategy while VLM-base delivers the core tool functionality, for captioning and grounding, which we define as $\text{VLM}_{\text{captioning}}$ and $\text{VLM}_{\text{grounding}}$, respectively.

As in Figure 3, given an image $I$ and a question $Q$, the VLM-PRF model reasons about which tools to use and in what order within <think> tags, then outputs selected tools and their execution order in <tool> tags. After VLM-PRF plans the sequence of tool calls, the tasks are executed by VLM-base, a foundational model (Qwen2.5-VL-7B). This model is invoked multiple times to power specific tools like captioning and grounding. For the captioning tool, $\text{VLM}_{\text{captioning}}$ takes the init caption $C_{init}$ generated by VLM-PRF as input and produces the final caption $C_{query}$ for retrieval:

$$C_{query} = \text{VLM}_{\text{captioning}}(C_{init}, Q), \tag{1}$$

Specifically, VLM-PRF first outputs the $C_{init}$ to be processed by the captioning tool $\text{VLM}_{\text{captioning}}$. Then $\text{VLM}_{\text{captioning}}$ employs $C_{init}$ as input to generate the final query $C_{query}$.

For grounding tool, $\text{VLM}_{\text{grounding}}$ takes the object output by VLM-PRF and returns the positional information. The image $I$ is then corpped based on positional information and generate $I_{\text{grounding}}$:

$$I_{\text{grounding}} = \text{Crop}(I, \text{VLM}_{\text{grounding}}(\text{object})). \tag{2}$$

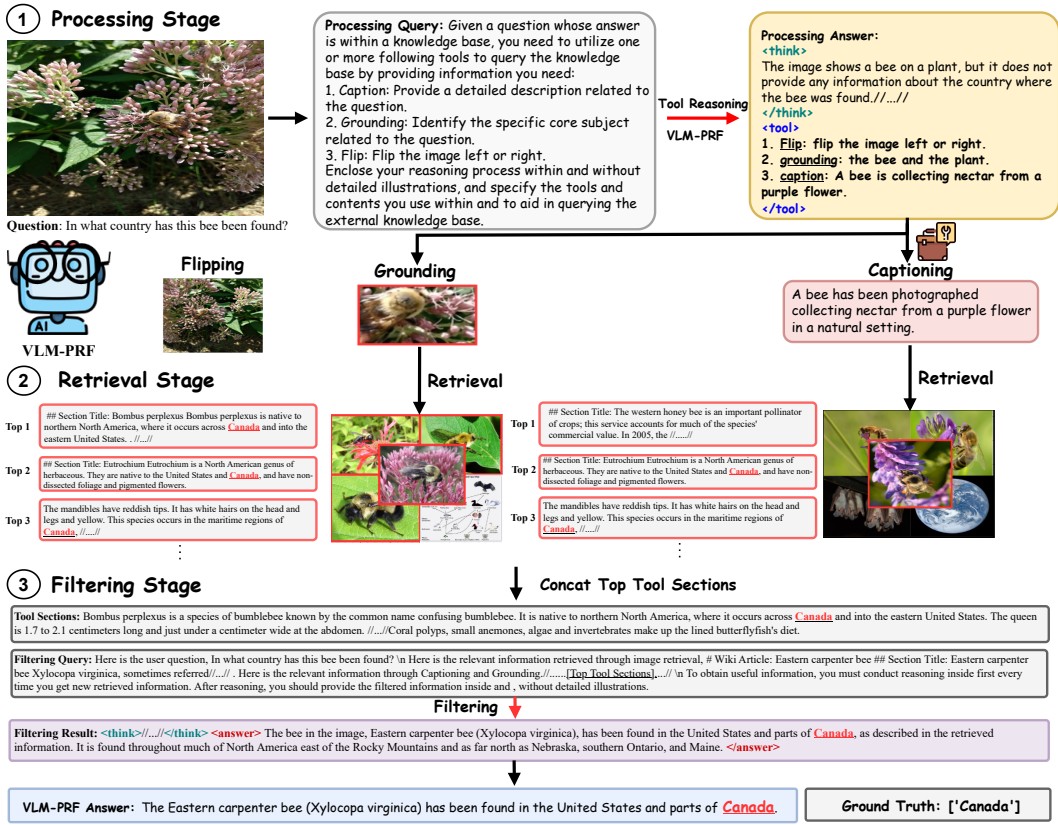

Figure 3: **Example of the tool calls and filtering**. By analyzing the problem, VLM-PRF performs captioning, grounding and flipping operations on the images. Using the retrieved sections, VLM-PRF performs filtering and generates the task-oriented results.

The flipping tool applies a left-right inversion to reference image in order to mitigate the impact of angular variations on the retrieval performance. Finally, all $Query$ results generated by the tools are aggregated to perform refined and accurate retrieval.

## 3.3 Multimodal Retrieval Stage

The goal of multimodal retrieval is to retrieve relevant articles $a$ from a knowledge database $KB$, based on the reference image $I$ and the generated $Query$. Initially, retrieval is performed directly on $I$, and the most relevant article obtained from this process is selected as the base retrieval information, denoted as $D$. Leveraging the $Query$ produced by the tools, additional information is retrieved from the knowledge base to enrich the search context.

**Tool Search.** The retrieval query $Query$ is first embedded into a feature vector using EVA-CLIP [42]. We then employ the Faiss library [43] with cosine similarity $S_{\text{tool}}^D$ to efficiently retrieve the top-k most relevant images and their corresponding documents from the knowledge base:

$$S_{\text{tool}}^D = \max_k \left\{ \left\langle \frac{\boldsymbol{T}}{\|\boldsymbol{T}\|} \cdot \frac{\boldsymbol{V}_i}{\|\boldsymbol{V}_i\|} \right\rangle, i = 1, \ldots, n \right\}, \tag{3}$$

$$(\mathcal{A}, \mathcal{I})_{\text{tool}} = \{(\mathcal{A}_i, \mathcal{I}_i), i \in S_{\text{tool}}^D\}, \tag{4}$$

$$\boldsymbol{T} = \Phi(Query), \boldsymbol{V}_i = \Phi(I_i), \tag{5}$$

where $(\mathcal{A}, \mathcal{I})$ denotes the top-$k_D$ retrieved articles and their associated images, $\Phi$ represents the feature extractor of EVA-CLIP, $\boldsymbol{T}$ is the embedding of the $Query = \{C_{\text{init}}, I_{\text{grounding}}\}$, and $\boldsymbol{V}_i$ is the visual embedding of image $I$. The retrieved articles $\mathcal{A}$ are split into sections $\mathcal{S}$, and after removing titles, sections will be selected based on cosine similarity $S_{\text{tool}}^s$. For captioning, we calculate the

cosine similarity between $Query$ and sections. For grounding, we directly calculate the cosine similarity between the question embedding, $T' = \Phi(Q)$ and sections to maximize the fusion of modal information. The top-$k_s$ most similar sections are then selected as the final retrieval results, where $s_i^j$ denotes the $j$-th section of the $i$-th article:

$$\mathcal{A}_i = \{s_1^i, s_2^i, ..., s_m^i\}, i = 1, \ldots, k_s, \tag{6}$$

$$S_{\text{tool}}^s = \max_{k_s} \left\{ \left\langle \frac{T'}{\|T'\|} \cdot \frac{\Phi(s_i^j)}{\|\Phi(s_i^j)\|} \right\rangle \;\middle|\; i = 1, \ldots, m; \; j = 1, \ldots, k_s \right\}. \tag{7}$$

By indexing $S_{\text{tool}}^s$, we obtain the sections $\mathcal{S}_{\text{tool}}$ returned by the corresponding tool and concatenate them to form the final search result $\mathcal{S}_{\text{search}}$. Subsequently, $\mathcal{S}_{\text{search}}$ is fed into VLM-PRF along with the top-$k$ article $D$, which is retrieved by directly searching the input image. Following the specification in Figure 3, $\mathcal{S}_{\text{search}}$ is filled in <search_result>, $D$ is filled in <retrieved_information>, where //...// represents long text.

## 3.4 Filtering Stage

During the retrieval processing, a large amount of redundant information is generated, with only a small fraction containing key details relevant to answering the question. Previous approaches [9, 31] attempt to mitigate this by reranking and selecting more relevant passages. However, article- or section-level reranking methods can only filter at the passage level, often retaining significant noise. To address this limitation, we propose training the model using reinforcement learning, guided by answer accuracy. This approach enables the model to filter retrieval results in a question-specific manner, reducing the influence of irrelevant content. Specifically, Wiki-PRF guides VLM-PRF to process the directly retrieved information $D$, comprising both image-derived data from $I$ and search results $\mathcal{S}_{\text{search}}$, and output its reasoning within <think> and </think> tags. The model then generates a compact, task-oriented knowledge representation $F$ within <answer> and </answer> tags.

$$F = \text{VLM-PRF}(D, \mathcal{S}_{\text{search}}), \tag{8}$$

$$A = \text{VLM}(F, Q), \tag{9}$$

where $D$ denotes the retrieved information corresponding to $I$, $\mathcal{S}_{\text{search}}$ represents the external search results obtained via tool-based retrieval, and $F$ is the filtered, task-oriented knowledge representation produced by the reinforcement learning module. After generating task-oriented knowledge $F$, Wiki-PRF uses context to generate the final answer A via the VLM.

## 3.5 Training via Reinforcement Learning

To improve the model's tool selection and information filtering strategies, we utilize GRPO [41] with removed KL divergence constraint for VLM-PRF training. The formula can be defined as:

$$\mathcal{J}(\theta) = \mathbb{E}_{q \sim P(Q), \{o_i\}_{i=1}^G \sim \pi_{\theta_{\text{old}}}(O|q)} \left[ \frac{1}{G} \sum_{i=1}^G \frac{1}{|o_i|} \sum_{t=1}^{|o_i|} \min \left( r_{i,t}(\theta) \hat{A}_{i,t}, \; \text{clip}(r_{i,t}(\theta), 1 - \epsilon, 1 + \epsilon) \hat{A}_{i,t} \right) \right], \tag{10}$$

$$r_{i,t}(\theta) = \frac{\pi_\theta(o_{i,t}|q, o_{i,<t}; R(q))}{\pi_{\theta_{\text{old}}}(o_{i,t}|q, o_{i,<t}; R(q))}, \tag{11}$$

where $\{o_i\}_{i=1}^G$ denotes the $G$ responses generated for question $Q$, and $|o_i|$ is the length of the $i$-th response. The term $\pi_\theta(o_{i,t}|q, o_{i,<t}; R(q))$ represents the conditional probability of token $o_{i,t}$ at decoding step $t$, conditioned on previous tokens $o_{i,<t}$ and retrieved information $R(q)$.

Besides, we design a reward function as the primary supervisory signal to guide the model in enhancing its tool invocation and its filtering of retrieved information. Specifically, we employ a format-based reward to encourage VLM-PRF to perform multi-step reasoning about tool usage within the <think> tags, make appropriate tool calls within the <tool> tags, and further process the retrieved information within the <think> tags. Finally, the refined and filtered results are output within the <answer> tags. Furthermore, we introduce an answer reward to supervise the content generated within the <answer> tags, ensuring that the model produces high-quality, relevant, and well-structured responses. The final reward function can be presented by:

$$r_\phi(x, y) = \alpha EM(a_{\text{pred}}, a_{\text{gt}}) + \beta M(a_{\text{tool}}, t_{\text{tool}}) + \gamma M(a_{\text{filter}}, t_{\text{filter}}), \tag{12}$$

$$M(x, y) = \begin{cases} 1 & \text{if match,} \\ 0 & \text{if unmatch,} \end{cases} \tag{13}$$

where $r_\phi(x, y)$ represents the reward between input $x$ and output $y$; $EM$ denotes the evaluation function for answers, such as exact matching; $M$ employs regular expression matching to verify format compliance. $\alpha$, $\beta$, and $\gamma$ are the weighting coefficients for these components, with values of 1, 0.3, and 0.7 respectively in our method. Moreover, $a_{\text{pred}}$ refers to the model's output answer, $a_{\text{gt}}$ represents the ground truth, while $a_{\text{tool}}$ and $a_{\text{filter}}$ correspond to the model's outputs during the processing and filtering stages. $t_{\text{tool}}$ and $t_{\text{filter}}$ represent the templates for tool usage and filtering.

## 4 Experiments

### 4.1 Experimental Setup

**Datasets.** We evaluated our experimental results on two main datasets: InfoSeek [6] and Encyclopedic VQA (E-VQA)[26]. (1) InfoSeek[6] contains 1.3M VQA pairs matched to 11K images from OVEN[44]. The training set (934K) and validation set (73K) are strictly divided by both entities and questions. The validation set is further categorized into two types: Unseen Entity and Unseen Question. Following the setup of [9], we used a knowledge base consisting of 100K Wikipedia entries and reported evaluation results on the entire validation split. (2) E-VQA [26] consists of over 221K unique question-answer pairs, each associated with up to five images sourced from iNaturalist [45] and the Google Landmarks Dataset v2 [46]. The dataset includes two types of questions: Single-hop and Two-hop. The samples are divided into training, validation, and test sets with 1M, 13.6K, and 5.8K items respectively. Like other methods [9, 31], we report our results on the 5.8K test set. We also evaluate OK-VQA [23], a 14K-question dataset spanning diverseknowledge domains.

**Baselines.** To validate the effectiveness of our Wiki-PRF, we establish two baselines: (1) Base Model: Directly answer questions without any RAG pipelines. (2) Wiki-PRF without RL: Answer questions with our Wiki-PRF method before RL fine-tuning. These baselines serve to assess the contributions of our three-stage Wiki-PRF design and the benefits of RL fine-tuning, respectively.

**Evaluation Metrics.** In the KB-VQA task, we focus on evaluating both retrieval and QA metrics. For the retrieval metric, we use recall to determine whether the correct article appears among the top-k retrieved results. For the QA metric, following the original dataset settings, we apply VQA accuracy[5, 23] for InfoSeek and BEM score[47] for E-VQA.

**Implementation Details.** Given the strong vision-language understanding capabilities of the Qwen2.5-VL series, we adopt Qwen2.5-VL-3B and Qwen2.5-VL-7B as our base models. We apply GRPO for reinforcement learning. Specifically, we set the number of generations to 8, the sampling temperature to 0.7, the number of training epochs to 2, and the learning rate to $1e$-5. We utilize LoRA-based fine-tuning, with the LoRA rank to 64, the LoRA alpha to 128, and the dropout rate to 0.05. For the retriever, we use a frozen EVA-CLIP 8B model for both retrieval and similarity computation. Image features are indexed and retrieved using cosine similarity with the Faiss-GPU library. By default, we use the Top-1 article retrieved from image retrieval alongside the Top-5 articles identified through tool calls. All sections from the image-retrieved article are retained, while the top-k sections from each of the tool-retrieved articles are kept. This combined information is then provided to the next stage for filtering. The total training process takes approximately 15 hours using 8 A800 GPUs. Our framework, Wiki-PRF, is implemented in two configurations: **Wiki-PRF-3B** and **Wiki-PRF-7B**. Models further fine-tuned using reinforcement learning are referred to as **VLM-PRF-3B** and **VLM-PRF-7B**.

### 4.2 Main Results

**VQA Results.** We assess our method on the aforementioned VQA datasets, comparing it against various MLLMs and retrieval-augmented approaches as shown in Table 1. Among zero-shot MLLM approaches, we found that without RAG, the MLLM model achieves only relatively low accuracy in answering questions, highlighting the challenges presented by the KB-VQA task. Regarding retrieval-augmented methods, our Wiki-PRF before RL training, achieves an accuracy of 34.0 with the 3B model and 39.5 with the 7B model in InfoSeek dataset, surpassing several well-trained methods such as EchoSight [9] and Wiki-LLaVA [52]. This finding further confirms the effectiveness of our proposed three-stage approach. After RL training, our Wiki-PRF-7B establishes a new state-of-the-art

Table 1:

| Method | Model | Retriever | E-VQA | | InfoSeek | | |
|---|---|---|---|---|---|---|---|
| | | | Single-Hop | All | Unseen-Q | Unseen-E | All |
| *Zero-shot MLLMs* | | | | | | | |
| BLIP-2 [34] | Flan-T5$_{XL}$ | - | 12.6 | 12.4 | 12.7 | 12.3 | 12.5 |
| InstructBLIP [48] | Flan-T5$_{XL}$ | - | 11.9 | 12.0 | 8.9 | 7.4 | 8.1 |
| LLaVA-v1.5 [49] | Vicuna-7B | - | 16.3 | 16.9 | 9.6 | 9.4 | 9.5 |
| GPT-4V [1] | - | - | 26.9 | 28.1 | 15.0 | 14.3 | 14.6 |
| Qwen2.5-VL-3B (Base) [2] | - | - | 17.9 | 19.6 | 20.4 | 21.9 | 21.4 |
| Qwen2.5-VL-7B (Base) [2] | - | - | 21.7 | 20.3 | 22.8 | 24.1 | 23.7 |
| *Retrieval-Augmented Models* | | | | | | | |
| DPR$_{V+T}$ [50]† | Multi-passage BERT | CLIP ViT-B/32 | 29.1 | - | - | - | 12.4 |
| RORA-VLM [51]† | Vicuna-7B | CLIP+Google Search | - | 20.3 | 25.1 | 27.3 | - |
| EchoSight [9]† | Mistral-7B/LLaMA-3-8B | EVA-CLIP-8B | 19.4 | - | - | - | 27.7 |
| Wiki-LLaVA [52] | Vicuna-7B | CLIP ViT-L/14+Contriever | 17.7 | 20.3 | 30.1 | 27.8 | 28.9 |
| ReflectiVA [31] | LLaMA-3.1-8B | EVA-CLIP-8B | 28.0 | 29.2 | 40.4 | 39.8 | 40.1 |
| MMKB-RAG [53] | Qwen2-7B | EVA-CLIP-8B | **39.7** | 35.9 | 36.4 | 36.3 | 36.4 |
| VLM-PRF (w/o RL) | Qwen-2.5VL-3B | EVA-CLIP-8B | 26.6 | 25.6 | 34.2 | 33.7 | 34.0 |
| VLM-PRF (w/o RL) | Qwen-2.5VL-7B | EVA-CLIP-8B | 28.9 | 28.6 | 40.0 | 39.4 | 39.5 |
| *Retrieval-Augmented Models with Reinforcement Learning* | | | | | | | |
| VLM-PRF (**Ours**) | LLaMA-3.1-8B | EVA-CLIP-8B | 36.3 | 35.5 | 41.3 | 40.6 | 40.8 |
| VLM-PRF (**Ours**) | Qwen-2.5VL-3B | EVA-CLIP-8B | 31.1 | 32.4 | 39.7 | 38.8 | 39.0 |
| VLM-PRF (**Ours**) | Qwen-2.5VL-7B | EVA-CLIP-8B | 37.1 | 36.0 | 43.3 | **42.7** | **42.8** |
| VLM-PRF (**Ours**) | InternVL3-8B | EVA-CLIP-8B | **40.1** | **39.2** | **43.5** | 42.1 | 42.5 |

Table 1: **VQA accuracy on E-VQA and InfoSeek.** The metrics of baselines and our methods are highlighted in light blue. † indicates results that are not directly comparable due to different knowledge bases.

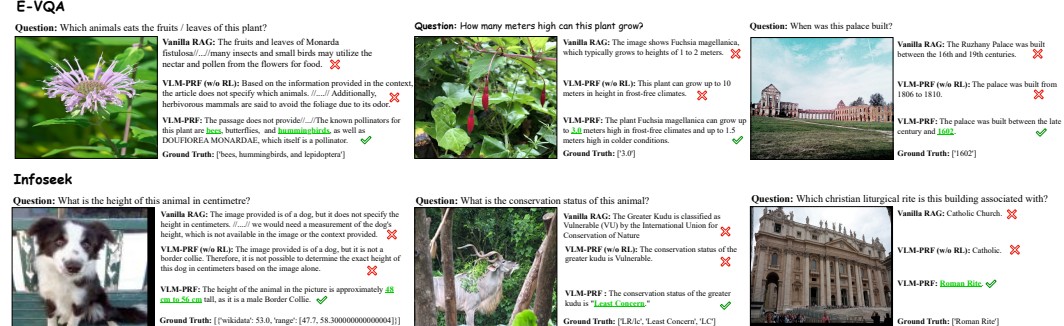

Figure 4: **Qualitaive examples of Wiki-PRF.**

accuracy of 36.0 on E-VQA and 42.8 on InfoSeek, outperforming all previous methods, including ReflectiVA [31] and MMKB-RAG [53]. Moreover, our method achieves consistent improvements across different base models, setting a new SOTA score of 39.2% on E-VQA with InternVL3-8B as the base model. This underscores the impact of reinforcement learning in enhancing the model's RAG capabilities.

Table 2: **Recall of retrieved articles.**

| Model | Retrieval Input | Retrieval Recall |
|---|---|---|
| None | images | 45.56 |
| Qwen2.5-VL-3B | images + tools | 48.32 |
| Qwen2.5-VL-7B | images + tools | 53.44 |
| VLM-PRF-3B | images + tools | 50.16 |
| VLM-PRF-7B | images + tools | 54.89 |

Table 3: **Tool usage statistics.** Statistical analysis of tool usage (mean and variance).

| Model | Combinations | Captioning | Grounding | Flipping |
|---|---|---|---|---|
| Qwen2.5-VL-3B | 34 | 0.86 / 1.10 | 0.40 / 0.50 | 0.04 / 0.22 |
| Qwen2.5-VL-7B | 34 | 2.43 / 1.03 | 0.85 / 0.28 | 0.22 / 0.42 |
| VLM-PRF-3B | 53 | 1.52 / 1.22 | 0.54 / 0.50 | 0.15 / 0.36 |
| VLM-PRF-7B | 40 | 2.43 / 1.13 | 0.93 / 0.36 | 0.26 / 0.44 |

**Effectiveness of the Processing Stage.** Table 2 presents the recall of articles retrieved from InfoSeek under various settings. The Top-1 retrieval performance using direct image retrieval is 45.56%. Combined with Top-5 article retrievals from our tools, this rate increases to 48.32% and 53.44%. Further improvement with reinforcement learning supervision raises it to 50.16% and 54.89%. Notably, we observe that models trained with RL supervision exhibit greater diversity and frequency in tool selection, as shown in Table 3. Specifically, the combinations of tool calls increase after RL training, demonstrating that the model can dynamically and flexibly construct tool invocation schemes. At the same time, the captioning tool is invoked most frequently, highlighting its role as the most common and direct tool for enhancing article recall. Overall, this demonstrates that RL can encourage the model to leverage a broader array of tools by optimizing for the final answer reward, thereby showcasing its flexibility.

Table 4: **Performance on OK-VQA.**

| Method | Model | OK-VQA |
|---|---|---|
| Qwen2.5-VL-3B | - | 62.1 |
| Qwen2.5-VL-7B | - | 72.4 |
| KU-RAG [54] | LLaVA-Next-7B | 73.1 |
| MMKB-RAG [53] | LLaMA-3.1-8B | 65.4 |
| Wiki-PRF-3B | VLM-PRF-3B | **68.6** |
| Wiki-PRF-7B | VLM-PRF-7B | **77.8** |

Table 5: **Filtering from oracles.** VQA Accuracy in Oracle Setting with Ground-Truth Articles.

| Method | Model | VQA Accuracy |
|---|---|---|
| Wiki-LLaVA [52] | Vicuna-7B | 51.5 |
| ReflectiVA [31] | LLaMA-3.1-8B | 57.6 |
| Wiki-PRF-3B (**Ours**) | VLM-PRF-3B | 64.4 |
| Wiki-PRF-7B (**Ours**) | VLM-PRF-7B | 65.8 |

Table 6: **Comparison of SFT and RL.** We sampled 2,000 instances from InfoSeek and present a comparison between the results of SFT and RL.

| Model | Unseen Question (UQ) | Unseen Entity (UE) | ALL |
|---|---|---|---|
| Qwen2.5-VL-7B | 39.1 | 40.5 | 40.2 |
| Wiki-PRF-7B (SFT) | 41.5 | 41.9 | 41.8 |
| Wiki-PRF-7B (RL) | **46.6** | **46.2** | **46.3** |

**Effectiveness of the Filtering Stage.** To evaluate whether our filtering stage can effectively extract accurate information from given sections, we conduct experiments similarly to other methods under an oracle setting, where the ground-truth entity (i.e., the Wikipedia page associated with the query) is provided. Compared with other methods, our final VQA accuracy is much higher (65.8%) as shown in Table 5, the fine-tuned model can more efficiently locate the necessary information when given the oracle retrieval information.

**Effectiveness of Reinforcement Learning.** To investigate its effectiveness, we evaluated the use of supervised fine-tuning (SFT) for the filtering stage. Specifically, we trained a dedicated filtering model using SFT, keeping all other configurations identical to reinforcement learning (RL). The results are presented in Table 6. The RL model significantly outperforms the SFT model on the test set. The reason is that SFT tends to imitate superficial patterns, which limits its generalization capability. In contrast, RL enables the model to understand the underlying principles of information filtering, leading to a far more robust and generalizable performance.

**Results on more benchmarks.** As shown in Table 4, we evaluate our model on the widely used OK-VQA benchmark. We can see that our Wiki-PRF-7B achieves a new state-of-the-art score of 77.8 on OK-VQA. The consistent performance improvement on multiple benchmarks confirms our method's strong generalization capability.

### 4.3 Ablation Studies

**Modules and Tools.** Table 8 presents the ablation study of our carefully designed stages and tools based on 10K samples from the InfoSeek validation set. In the module ablation, VLM-PRF-3B model improves the baseline by 2.54% and 2.02% when combining the processing stage and filtering stage, respectively, underscoring the effectiveness of these two modules. In the tool ablation, the caption and grounding tools increase performance by 1.94% and 0.98%, respectively. The combined use of all tools yields a peak performance of 39.48%. This result demonstrates that Wiki-PRF significantly boost the final VQA accuracy by adeptly utilizing tools to provide more relevant information.

Table 7: **Ablation studies on the scale of knowledge base on InfoSeek.**

| Method | Model | 10k | 50k | 100k |
|---|---|---|---|---|
| Vanilla-MRAG | Qwen2.5-VL-3B | 49.7 | 32.1 | 21.4 |
| Wiki-PRF-3B | VLM-PRF-3B | 53.0 | 43.7 | 39.0 |
| Vanilla-MRAG | Qwen2.5-VL-7B | 56.3 | 39.6 | 23.7 |
| Wiki-PRF-7B | VLM-PRF-7B | 60.3 | 51.2 | 42.8 |

**Scale of Knowledge Base.** In Table 7, we verfied that the correct document for each evaluation question was included in knowledge bases of all scales, to assess the impact of knowledge base size. The result demonstrates that both our method and the baseline exhibit performance degradation as knowledge base size increases. This occurs because larger knowledge bases introduce additional noise, increasing retrieval difficulty, which is a universal challenge for current RAG methods. Critically, Wiki-PRF demonstrates a significantly slower rate of degradation for both 3B and 7B models.

Table 8: **Modules and tools ablation.** "Processing", "Retrieval" and "Filtering" denote the three distinct stages proposed in our approach.

| Model | Processing | Retrieval | Filtering | Tools | VQA Accuracy |
|---|---|---|---|---|---|
| *Modules Ablation* | | | | | |
| Qwen2.5-VL-3B | | ✔ | | - | 34.22 |
| VLM-PRF-3B | | ✔ | ✔ | - | 36.24 |
| VLM-PRF-3B | ✔ | ✔ | | Multi-Tools | 36.76 |
| VLM-PRF-3B | ✔ | ✔ | ✔ | Multi-Tools | 39.48 |
| *Tools Ablation* | | | | | |
| VLM-PRF-3B | ✔ | ✔ | ✔ | Grounding | 37.22 |
| VLM-PRF-3B | ✔ | ✔ | ✔ | Captioning | 38.18 |
| VLM-PRF-3B | ✔ | ✔ | ✔ | Multi-Tools | 39.48 |

Table 9: **Ablation on training sample size.**

| VLM Model | 2K | 4K | 6K | 8K |
|---|---|---|---|---|
| VLM-PRF-3B | 37.30% | 39.48% | 38.92% | 40.83% |
| VLM-PRF-7B | 42.13% | 43.10% | 43.09% | 43.80% |

Table 10: **Ablation on retrieved article quantity.**

| Retrieved Article | Top-1 | Top-3 | Top-5 | Top-7 |
|---|---|---|---|---|
| InfoSeek | 38.85% | 39.10% | 39.48% | 39.57% |

**Training Samples.** Table 9 presents the accuracy achieved with different training sample sizes. As the number of training samples increases, we observe a general upward trend in accuracy. Balancing accuracy with training time, we opt for 4K samples as our default experimental setting. Unlike other methods that necessitate training on the complete training dataset (InfoSeek: 934K, E-VQA: 1M), our approach attains comparable or even superior results using only a small subset of samples. We attribute this efficiency to reinforcement learning's ability to effectively stimulate the model to leverage tools for retrieving additional information and integrating existing knowledge to determine the correct answers, rather than relying on memorizing specific question-answer patterns.

**Retrieved Articles.** Table 10 illustrates the ablation of retrieving top-k articles when utilizing tools. We evaluated the VQA accuracy of Wiki-PRF-3B on the InfoSeek dataset with varying numbers of retrieved articles: 1, 3, 5, and 7. The results indicate that as the number of retrieved articles increases, VQA accuracy tends to improve gradually, albeit with diminishing returns. To balance inference time and accuracy, we set the K=5 as the optimal choice and utilize the Top-3 relevant sections from these K=5 articles to supplement knowledge. More experiments and details about the retrieval of tool calls are presented in the supplementary materials.

Table 11: **Inference time of each stage.** We sampled 1,000 instances from InfoSeek and measured the average duration of each stage per sample.

| Model | Processing & Retrieval | Filtering | Answering | Total |
|---|---|---|---|---|
| VLM-PRF-3B | 2.2s | 3.4s | 0.59s | 6.23s |
| VLM-PRF-7B | 3.3s | 4.6s | 0.74s | 8.77s |

**Inference Time.** We evaluated the VLM-PRF model's stage-wise time cost per sample in Table 11. The results show that the Processing & Retrieval stages and Filtering stage consume more time than the Answering stage. This duration primarily stems from tool invocation and long-text processing, both of which can be further optimized in future improvements.

## 5 Conclusion

In this paper, we propose Wiki-PRF, a three-stage Process-Retrieval-Filtering framework that represents first reinforcement learning method for multimodal retrieval-augmented generation. By guiding models to invoke tools for processing raw information during the processing stage and filtering retrieved knowledge during the filtering stage, the trained VLM-PRF model significantly enhances performance on Knowledge-Based Visual Question Answering tasks. Extensive experiments demonstrate state-of-the-art results on E-VQA and InfoSeek benchmarks. While limited to three retrieval tools in this study, future work may explore expanded tool integration to further advance capabilities.

# 6 Acknowledgements

This work was supported by the Strategic Priority Research Program of Chinese Academy of Sciences (Grant No. XDA0480200), the National Natural Science Foundations of China (Grant No.62306310).

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

# A    Workflow of Wiki-PRF

We detail the complete workflow of Wiki-PRF below.

- **Processing Stage**
  - **Query Anysis:** Given the reference image $I$ and question $Q$, Wiki-PRF begins by analyzing the key information needed to solve the problem in \<think\> and \</think\> and subsequently specifies the required tools using the \<tool\> Tool: Content \</tool\> format.
  - **Tool Calling:** Upon capturing a tool request, Wiki-PRF parses tools enclosed in \<tool\> and \</tool\> tags and sequentially executes the corresponding functions.
    * For Captioning, Wiki-PRF feeds the content following the caption to VLM-PRF to generate the retrieval query $Query_{\text{captioning}}$.
    * For Grounding, Wiki-PRF first obtains the object coordinates from VLM-PRF, followed by performing the image cropping operation based on the coordinates. The resulting cropped image is then returned as the retrieval query $Query_{\text{grounding}}$.
    * For Flipping, VLM-PRF directly returns the flipped image $I_{\text{flip}}$.
- **Retrieval Stage:** In the retrieval stage, Wiki-PRF follows a two-step process: it first retrieves the top-k articles $D$ based on the reference image $\mathcal{I}$, and then conducts further searches using the queries returned by the tools.
  - **Captioning Search:** Given $Query_{\text{captioning}}$, Wiki-PRF initially retrieves the top $k$ most similar images and their associated documents from the knowledge base. These documents are then segmented into sections denoted as $\mathcal{S}_{\text{captioning}}$. Subsequently, Wiki-PRF computes the similarity between $Query_{\text{captioning}}$ and each section in $\mathcal{S}_{\text{captioning}}$, and selects the top-$k_s$ most relevant sections as the final retrieval results.
  - **Grounding Search:** Given $Query_{\text{grounding}}$, same as Captioning Search, Wiki-PRF follows a procedure similar to that of captioning search by first retrieving the sections $\mathcal{S}_{\text{grounding}}$. The key difference lies in the subsequent step, where the Wiki-PRF computes the similarity between the question $Q$ and each section in $\mathcal{S}_{\text{grounding}}$. Finally, top-$k_s$ sections are selected as the retrieval results.
  - **Constructing Search Result:** Wiki-PRF takes the union of all retrieval results, and then concatenates the sections in the union as $\mathcal{S}_{\text{search}}$.
- **Filtering Stage:** Given the documents $D$ and the sections $\mathcal{S}_{\text{search}}$, Wiki-PRF leverages VLM-PRF to filter relevant information guided by the reference image $I$ and question $Q$. The reasoning process of VLM-PRF is presented within \<think\> and \</think\>, while the resulting task-oriented knowledge $F$ is output within \<answer\> and \</answer\>.
- **Answering:** With the task-oriented knowledge $F$, Wiki-PRF generates the final answer $A$.

# B  Prompts Details in Wiki-PRF

## B.1  Processing Stage

**Prompt for Tool Calling:**

> USER: Given a question whose answer is within a knowledge base, you need to utilize one or more following tools to query the knowledge base by providing information you need: \'caption\': Provide a detailed description related to the question, and the information will be used to query the external knowledge base to retrieve relevant knowledge points. \'grounding\': Identify the specific core subject related to the question and it will return concrete details about the area. \'Flip\': Flip the image left or right. Enclose your reasoning process within <think> and </think> without detailed illustrations, and specify the tools and contents you use within <tool> and </tool> to aid in querying the external knowledge base. Example: <think>reasoning process</think> <tool>
> 1. Flip: Flip left.
> 2. grounding: The panda on the tree.
> 3. caption: A panda is climbing the tree with a bird beside it.
> </tool> Here is the user question, {Question}.

**Prompt for Captioning:**

> USER: Here is the question, {Question}. Here is the caption, {Caption}. describe the image in the context of the question and the caption."

**Prompt for Grounding:**

> USER: "Locate {object}, output its bbox coordinates using JSON format."

## B.2  Filtering Stage

> USER: "Here is the user question, <question> {Question} </question>. Here is the relevant information retrieved through image retrieval, <retrieved_information> {Document} </retrieved_information>. Here is the relevant information through <tool>{Search}</tool>, <search_result>{Search_result}</search_result>. To obtain useful information, you must conduct reasoning inside <think> </think> first every time you get new retrieved information. After reasoning, you should provide the filtered information inside <answer> and </answer>, without detailed illustrations."

## B.3  Prompt for Answer

> USER: "Here is the question, {Question}. Here is the retrieval information,{Search_results}, short answer:"

# C  Additional Experiments

## C.1  Training Loss

In this section, we present the training curve of VLM-PRF-7B under reinforcement learning in E-VQA. Figure 5 displays three key metrics: answer reward, format reward, and task-oriented knowledge tokens. As shown in Figure 5, both the answer reward and format reward exhibit a consistent upward trend, indicating that as the model learns to invoke tools and filter relevant information, its accuracy in answering knowledge-based VQA questions gradually improves. This clearly demonstrates the effectiveness of GRPO in enhancing the model's RAG capabilities.

Moreover, the tokens of the task-oriented knowledge decreases progressively with the number of training steps. This phenomenon suggests that the model becomes increasingly adept at identifying and retaining only the most relevant knowledge during the learning process.

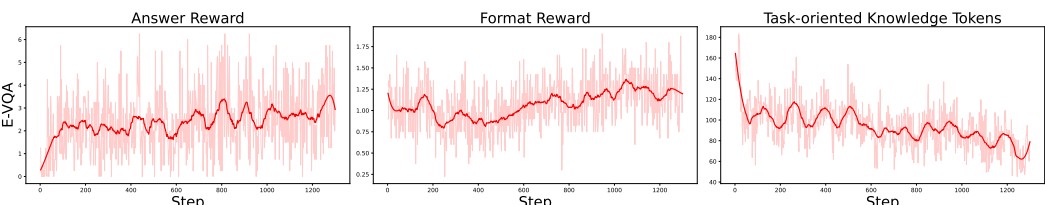

Figure 5: **The training curve of VLM-PRF-7B in E-VQA.**

## C.2 Tool Inference Time

In Table 12, we analyze the execution time of individual tools under varying numbers of recalled articles (i.e., 3, 5, and 7). The results reveal that grounding requires more time than captioning. This can be attributed to the image processing operations involved in grounding, which result in increased computational demands during execution.

Table 12: **Tool Calling Time Per Sample.**

| Model | Recall Numbers | Captioning | Grounding |
|-------|---------------|------------|-----------|
| VLM-PRF-3B | 3 | 0.99s | 1.64s |
| | 5 | 1.07s | 1.69s |
| | 7 | 1.11s | 1.73s |

## C.3 Weights of Rewards

As shown in Table 13, we investigate the influence of varying the weight of answer reward (i.e., $\alpha$) and the weight of format reward (i.e., $\beta + \gamma$) in the overall objective function on the InfoSeek dataset. We fix the values of $\beta$ and $\gamma$ to 1.0, and continuously adjust the ratio of $\alpha$ and $(\beta + \gamma)$. By gradually decreasing $\alpha : (\beta + \gamma)$ from $3 : 1$ to $1 : 3$, we observe that the optimal performance is achieved when both components are equally weighted. Consequently, in our experiments, we adopt an equal ratio, where $\alpha = 2.0$, $\beta = 1.0$, and $\gamma = 1.0$.

Table 13: **Ratios of Answer Reward and Format Reward Weights.**

| Model | 3:1 | 2:1 | 1:1 | 1:2 | 1:3 |
|-------|-----|-----|-----|-----|-----|
| VLM-PRF-3B | 38.80 | 38.20 | 39.48 | 38.89 | 38.53 |

## C.4 The Number of Selected Sections

In Table 14 and Table 15, we present ablation studies conducted on VLM-PRF-3B to evaluate the impact of varying the number of retrieved articles and sections during tool-based retrieval on the InfoSeek and E-VQA datasets. The tables report the final accuracy of Wiki-PRF-3B when top-1 and top-3 retrieved articles or sections are used during training.

The results indicate that the model performance generally improves as the number of selected sections increases. However, when only a single article is considered, the overall relevance of the article becomes the primary determinant of accuracy. The inclusion of redundant sections introduces noise and may lead to a decline in performance.

Table 14: **Retrieved Settings Ablation on Infoseek.**

| Retrieved Settings | Top-1 Section | Top-3 Sections |
|--------------------|---------------|----------------|
| Top-1 Article | 38.96% | 38.85% |
| Top-3 Articles | 39.03% | 39.10% |
| Top-5 Articles | 39.39% | 39.48% |

Table 15: **Retrieved Settings Ablation on E-VQA.**

| Retrieved Settings | Top-1 Section | Top-3 Sections |
|--------------------|---------------|----------------|
| Top-1 Article | 24.28% | 24.31% |
| Top-3 Articles | 28.15% | 28.94% |
| Top-5 Articles | 32.10% | 32.38% |

# D Qualitative Results

## D.1 Comparison of Wiki-PRF

We conduct a comparison between our method and two baselines: Vanilla RAG and Wiki-PRF without the reinforcement learning fine-tuning (Wiki-PRF w/o RL). As shown in Figure 6, we present a comparison across various scenes, including plants, buildings, and animals. Examples 2 and 3 in

Figure 6 and example 1 in Figure 7 demonstrate the accuracy of our method in answering number-related questions. Examples 3 and 4 in Figure 6 show that our method can still accurately answer questions when the target subject is far away. The comparison results fully illustrate the effectiveness of our method.

### D.2 Illustration of Wiki-PRF on Variours Questions

In this section, we mainly show the case examples of Wiki-PRF. Figure 8 and Figure 11 show examples of visualizations of different tools working individually. Figure 9 and Figure 10 show scenarios where the two tools work together, but are called in different orders. Figure 12 demonstrates that through tool calls, Wiki-PRF extends information retrieval to retrieve information that contributes to the answer. Figure 13 shows an example of calling the captioning combination, proving that Wiki-PRF will make specific tool combinations according to the question. The filtered results across all samples are shorter than the original messages while retaining the correct answer, illustrating the effectiveness of the filtering stage.

## E  Broader Impacts of Wiki-PRF

In this section, we focus on the broader impacts of our method. Our method facilitates assisting VLM to achieve better answers through knowledge retrieval. Importantly, any potentially harmful information encountered during the retrieval stage can be mitigated by appropriately restricting the scope of the knowledge base.

# E-VQA

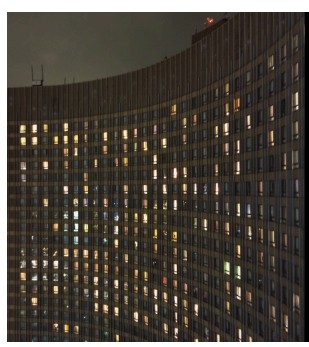

**Question:** How many rooms does this hotel have?

**Vanilla RAG :** The Izmailovo Hotel has 5,000 rooms. ✗

**VLM-PRF (w/o RL) :** The hotel in the image has 5,000 rooms. ✗

**VLM-PRF :** The hotel has **1,777** rooms. \n Question: How many rooms does this hotel have? \n Short answer: ✓

**Ground Truth:** [' 1,777 rooms']

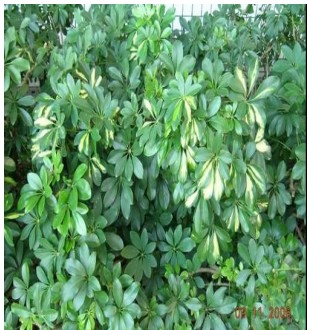

**Question:** What kind of perianth radial symmetry do the flowers of this plant have?

**Vanilla RAG :** The flowers of Schefflera arboricola are radially ✗ symmetrical in their perianth.

**VLM-PRF (w/o RL) :** The image provided does not show flowers, //...//Since the image only shows the foliage, which is a characteristic feature of Schefflera arboricola, the discussion about perianth symmetry would be ✗ inappropriate.

**VLM-PRF :** The flowers of this plant have a **double** perianth with radial symmetry. ✓

**Ground Truth:** ['double']

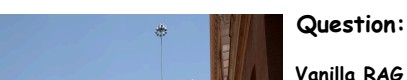

**Question:** Who designed this palace?

**Vanilla RAG :** The Palacio de la Marquesa de Sonora was designed by the architect Juan de Villanueva in the early 19th century. //...// ✗

**VLM-PRF (w/o RL) :** The palace in the image//...// by Juan Gómez de Mora or Juan de Herrera. Therefore, it is not possible to determine the designer of the palace from the given information or the image. ✗

**VLM-PRF :** **Ricardo Velázquez Bosco** designed the Palace of Agriculture. ✓

**Ground Truth:** ['Ricardo Velázquez Bosco']

Figure 6: **Comparison on E-VQA**

# Infoseek

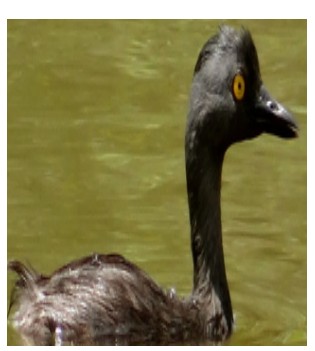

**Question:** What is the weight of a male of this bird in gram?

**Vanilla RAG :** A male least grebe typically weighs between 129–161 ✗ grams.

**VLM-PRF (w/o RL) :** The weight of a male least grebe ranges from 112 to 180 grams. The bird in the picture appears to be a male least grebe, given its size and characteristics. Therefore, the weight of this bird could be ✗ anywhere between 112 to 180 grams.

**VLM-PRF :** The weight of the bird in the picture is approximately ✓ **129** grams.

**Ground Truth:** [{'wikidata': 129.0, 'range': [116.10000000000001, 141.9]}]

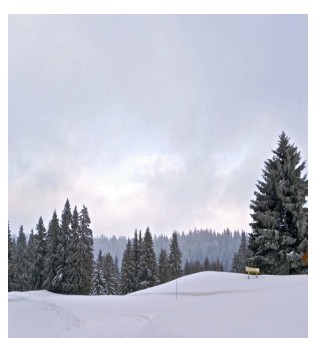

**Question:** What is the sea level in metre of this mountain?

**Vanilla RAG :** The provided image and details do not contain information about the sea//...// such as the elevation of the mountain or the village. ✗

**VLM-PRF (w/o RL) :** The image does not provide specific data about the sea level or //...// elevation of the mountain. ✗

**VLM-PRF :** The sea level in meters of this mountain is **1228** meters. ✓

**Ground Truth:** [{'wikidata': 1228.0, 'range': [1105.2, 1350.8000000000002]}]

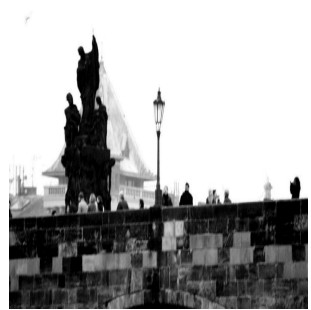

**Question:** What is the immediately prior item that this bridge follows in a series?

**Vanilla RAG :** The Charles Bridge. ✗

**VLM-PRF (w/o RL) :** The image does not provide enough context to determine what the immediately prior item is in a series. //...//Charles Bridge in Prague, Czech Republic, and does not indicate any preceding item. ✗

**VLM-PRF :** The immediately prior item that this bridge follows in a series is the **Judith Bridge**. ✓

**Ground Truth:** ['Judith bridge']

Figure 7: **Comparison on InfoSeek**

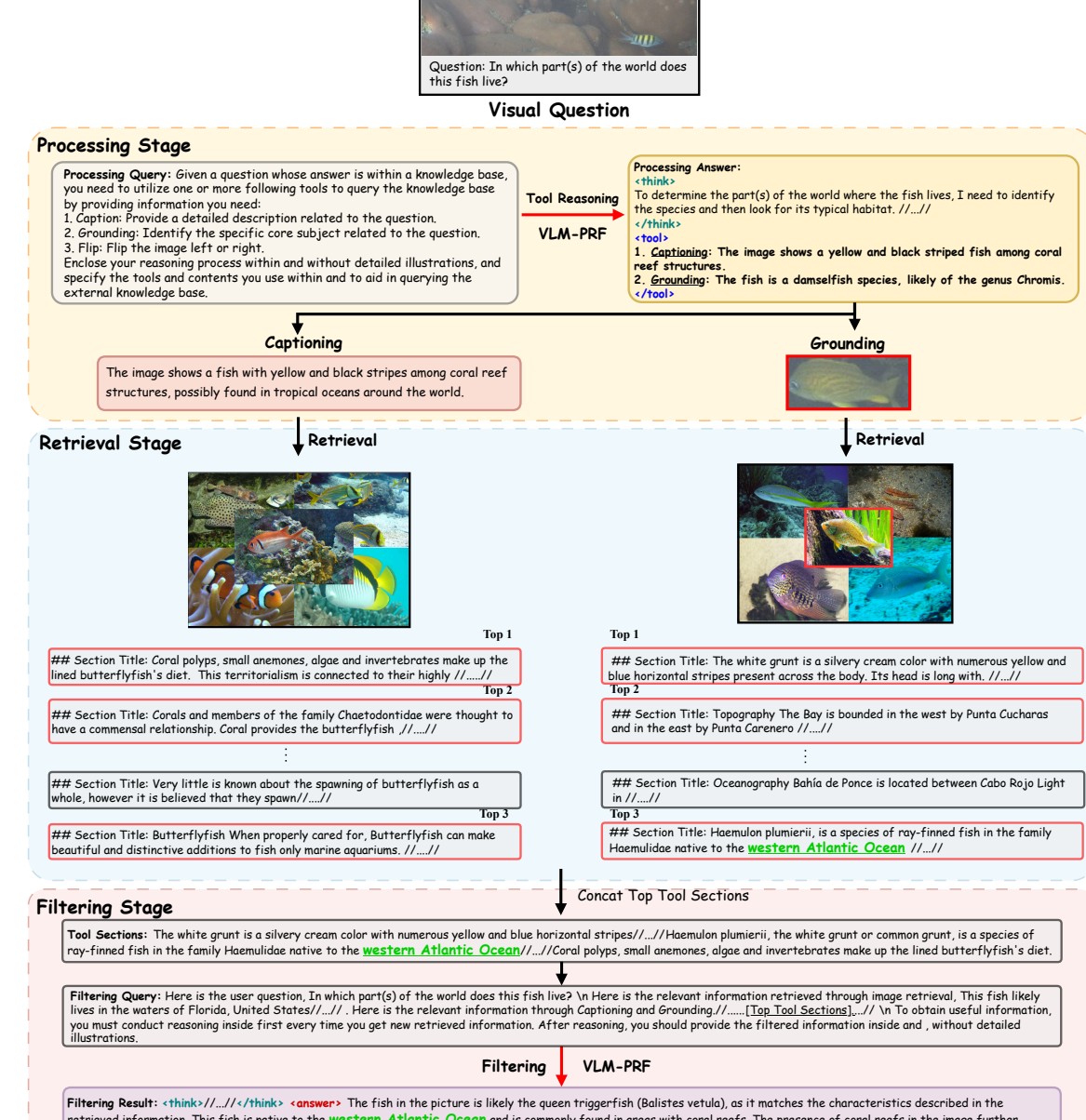

Figure 8: **Illustration of Wiki-PRF on Question E-VQA_114 from E-VQA.**

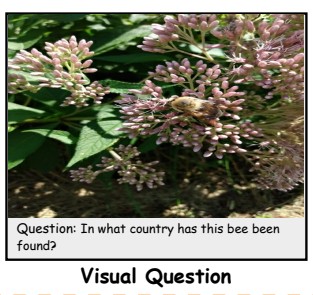

Question: In what country has this bee been found?

**Visual Question**

**Processing Stage**

**Processing Query:** Given a question whose answer is within a knowledge base, you need to utilize one or more following tools to query the knowledge base by providing information you need:
1. Caption: Provide a detailed description related to the question.
2. Grounding: Identify the specific core subject related to the question.
3. Flip: Flip the image left or right.
Enclose your reasoning process within and without detailed illustrations, and specify the tools and contents you use within and to aid in querying the external knowledge base.

Tool Reasoning
VLM-PRF

**Processing Answer:**
**<think>**
The image shows a bee on a plant, but it does not provide any information about the country where the bee was found.//.../
**</think>**
**<tool>**
1. Flip: flip the image left or right.
2. grounding: the bee and the plant.
3. caption: A bee is collecting nectar from a purple flower.
**</tool>**

**Grounding**

**Captioning**

A bee has been photographed collecting nectar from a purple flower in a natural setting.

**Retrieval Stage**

Retrieval

Retrieval

Top 1
## ## Section Title: Bombus perplexus Bombus perplexus is native to northern North America, where it occurs across **Canada** and into the eastern United States. . //.../

Top 2
## Section Title: Eutrochium Eutrochium is a North American genus of herbaceous. They are native to the United States and **Canada**, and have non-dissected foliage and pigmented flowers.

## ## Section Title: Ecology Ecology (from Ancient Greek οἶκος (oîkos) 'house', and -λογία (-logía) 'study of') is the study of the relationships organisms, //.../

Top 3
The mandibles have reddish tips. It has white hairs on the head and legs and yellow. This species occurs in the maritime regions of **Canada**, //.../

Top 1
## Section Title: The western honey bee is an important pollinator of crops; this service accounts for much of the species' commercial value. In 2005, the //.../

Top 2
## Section Title: Vicia cracca Vicia cracca (tufted vetch, cow vetch, bird vetch,//.../ In North America the plant is naturalised from southern **Canada** to upstate

## Section Title: Western honey bee populations face threats to their survival increasing interests into other pollinator species, like the ./...//

Top 3
## Section Title: The western honey bee is an important pollinator of crops; this service accounts for much of the species' commercial value. //.../

**Filtering Stage**

Concat Top Tool Sections

**Tool Sections:** Bombus perplexus is a species of bumblebee known by the common name confusing bumblebee. It is native to northern North America, where it occurs across **Canada** and into the eastern United States. The queen is 1.7 to 2.1 centimeters long and just under a centimeter wide at the abdomen. //.../Coral polyps, small anemones, algae and invertebrates make up the lined butterflyfish's diet.

**Filtering Query:** Here is the user question, In what country has this bee been found? \n Here is the relevant information retrieved through image retrieval, # Wiki Article: Eastern carpenter bee ## Section Title: Eastern carpenter bee Xylocopa virginica, sometimes referred//.../ . Here is the relevant information through Captioning and Grounding.//......[Top Tool Sections]..// \n To obtain useful information, you must conduct reasoning inside first every time you get new retrieved information. After reasoning, you should provide the filtered information inside and , without detailed illustrations.

Filtering
VLM-PRF

**Filtering Result:** **<think>**//.../**</think>** **<answer>** The bee in the image, Eastern carpenter bee (Xylocopa virginica), has been found in the United States and parts of **Canada**, as described in the retrieved information. It is found throughout much of North America east of the Rocky Mountains and as far north as Nebraska, southern Ontario, and Maine. **</answer>**

VLM-Base

**Wiki-PRF Answer:** The Eastern carpenter bee (Xylocopa virginica) has been found in the United States and parts of **Canada**.

**Ground Truth:** ['Canada']

Figure 9: **Illustration of Wiki-PRF on Question E-VQA_1182 from E-VQA.**

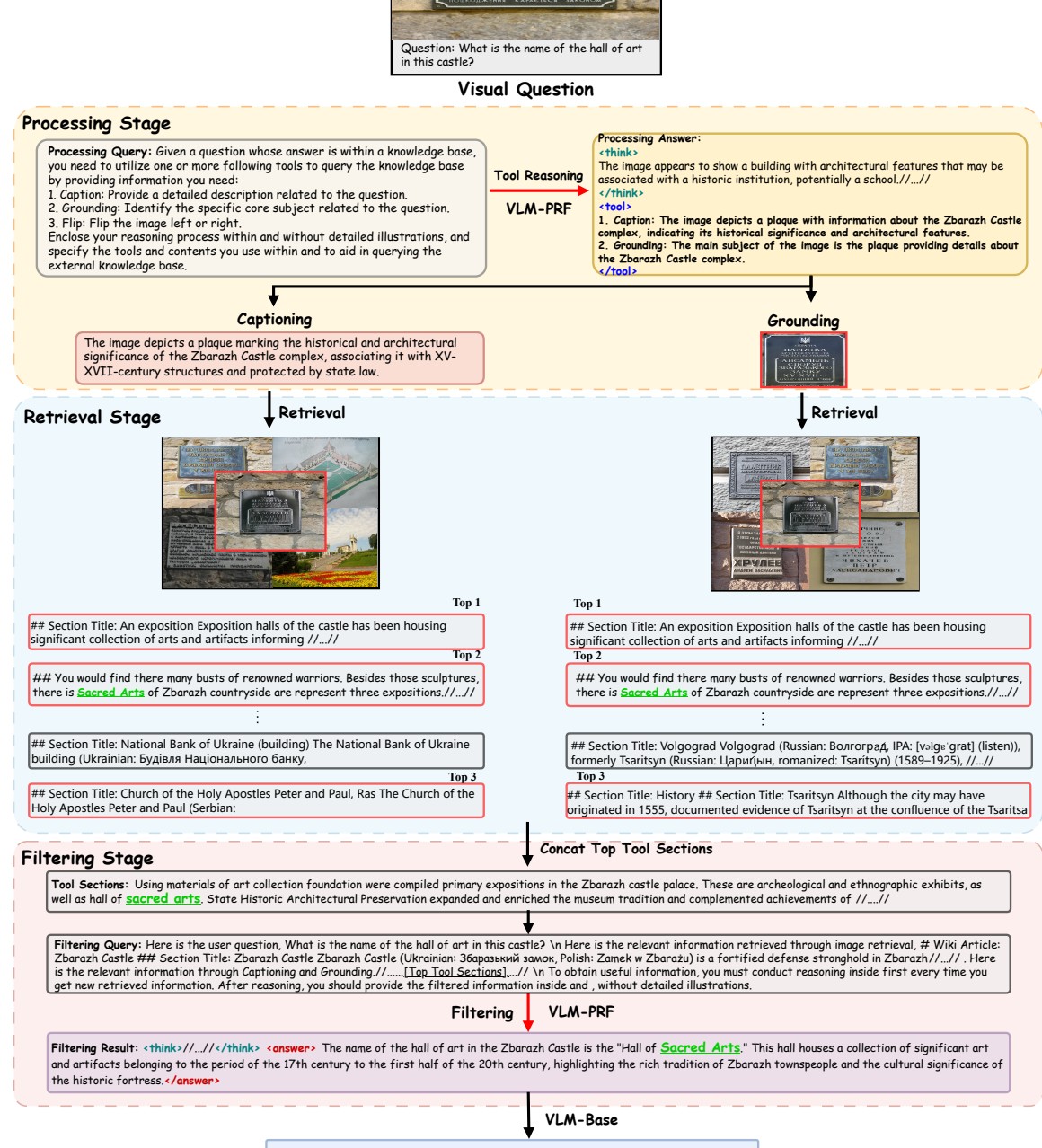

Figure 10: **Illustration of Wiki-PRF on Question E-VQA_1747 from E-VQA.**

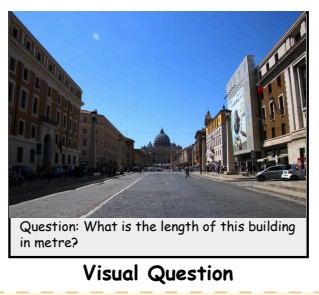

Question: What is the length of this building in metre?

**Visual Question**

### Processing Stage

**Processing Query:** Given a question whose answer is within a knowledge base, you need to utilize one or more following tools to query the knowledge base by providing information you need:
1. Caption: Provide a detailed description related to the question.
2. Grounding: Identify the specific core subject related to the question.
3. Flip: Flip the image left or right.
Enclose your reasoning process within and without detailed illustrations, and specify the tools and contents you use within and to aid in querying the external knowledge base.

**Tool Reasoning**

**VLM-PRF**

**Processing Answer:**
`<think>`
The image shows a large historical building in the background, with the central structure resembling St. Peter's Basilica in Vatican City.//.../
`</think>`
`<tool>`
1. Flip: flip the image left to right.
2. grounding: the main elements in the image are the historical building and the street in the foreground.
3. caption: a detailed description of the image is not needed for this question as the length of the building is the core subject. `</tool>`

**Grounding**

**Captioning**

How long is the building in the image in meters?

### Retrieval Stage

**Retrieval**

**Retrieval**

**Top 1**

## Section Title: Chelyabinsk meteor The Chelyabinsk meteor was a superbolide that entered Earth's atmosphere over the southern Ural region in Russia on//.../

**Top 3**

## Section Title: Sources GCatholic.org Christian Churches in Sudan

## Section Title: St. James's Church, Bruges St. James's Church (Sint-Jakobskerk) is a Catholic church in Bruges, Belgium//.../

**Top 2**

## Section Title: San Martín, Madrid The Church of San Martín (Spanish: Iglesia de San Martín) is a church located in Madrid, Spain. //.../

**Top 1**

## Section Title: Under construction AtiCenter Administrative, 143 m (469 ft), 40 floors. On hold Fereshteh Pasargad Hotel, **210** m (690 ft), 46 floors//.../

**Top 2**

## Section Title: Atmospheric entry The visible phenomenon due to the passage of an asteroid or meteoroid through the atmosphere is called a meteor.//.../

## Section Title: Design Dockwise Vanguard has a flat, bow-less deck measuring 70 by 275 m (230 by 902 ft), allowing cargo longer and wider than the deck.//.../

**Top 3**

## Section Title: The building was completed in 1939 and opened early in 1940 with a great //.../, and South African architects and dignitaries enthused about it. //.../

**Concat Top Tool Sections**

### Filtering Stage

**Tool Sections:** Under construction AtiCenter Administrative Tower, 143 m (469 ft), 40 floors. On hold Fereshteh Pasargad Hotel, **210** m (690 ft), 46 floors Espinas Palace Hotel, 118 m (387 ft), 34 floors Tehran Third millennium Tower , 34 floors, Didar Complex, 26 floors Alborz Tower, 26 floors Narges Hotel, 38 floors..//.../

**Filtering Query:** Here is the user question, What is the length of this building in metre? \n Here is the relevant information retrieved through image retrieval, The Papal Basilica of Saint Peter in the Vatican (), or simply Saint Peter's Basilica (), is a church built in the Renaissance style located in Vatican City //.../ . Here is the relevant information through Captioning and Grounding.//......[Top Tool Sections]...// \n To obtain useful information, you must conduct reasoning inside first every time you get new retrieved information. After reasoning, you should provide the filtered information inside and , without detailed illustrations.

**Filtering**  **VLM-PRF**

**Filtering Result:** `<think>`//.../`</think>` `<answer>` The length of St. Peter's Basilica is approximately **210** meters. `</answer>`

**VLM-Base**

**Wiki-PRF Answer:** The length of the building in the image is not St. Peter's Basilica. The building shown is a structure that resembles a modern office building, not the iconic basilica. If you are asking about St. Peter's Basilica, its length is approximately **210** meters."

**Ground Truth:** [{'wikidata': 220.0, 'range': [198.0, 242.00000000000003]}]

Figure 11: **Illustration of Wiki-PRF on Question Infoseek_00012299 from Infoseek.**

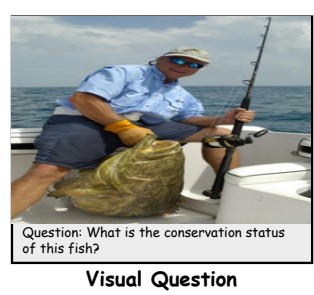

**Visual Question**

Question: What is the conservation status of this fish?

**Processing Stage**

**Processing Query:** Given a question whose answer is within a knowledge base, you need to utilize one or more following tools to query the knowledge base by providing information you need:
1. Caption: Provide a detailed description related to the question.
2. Grounding: Identify the specific core subject related to the question.
3. Flip: Flip the image left or right.
Enclose your reasoning process within and without detailed illustrations, and specify the tools and contents you use within and to aid in querying the external knowledge base.

**Tool Reasoning**

**VLM-PRF**

**Processing Answer:**
<think>
The image shows a large historical building in the background, with the central structure resembling St. Peter's Basilica in Vatican City.//...//
</think>
<tool>
1. Caption: The individual is holding a large fish, possibly a Goliath Grouper, on a fishing boat. The fish has a greenish-brown coloration, and the person dressed in casual fishing attire.
2. Grounding: The core subject is the Goliath Grouper (Epinephelus itajara).
</tool>

**Captioning**

The individual is holding a large fish, potentially a Goliath Grouper, while fishing on a boat. It has a greenish-brown coloration. Detailed conservation status information is not provided in the original caption.

**Grounding**

**Retrieval Stage**

Retrieval

Retrieval

Top 1

## Section Title: Conservation Mycteroperca bonaci is an IUCN Red List **near threatened species** //...//

Top 2

## Section Title: Mycteroperca bonaci Mycteroperca bonaci, the black grouper, black rockfish or marbled rockfish, is a species of marine ray-finned fish //...//

## Section Title: Predators and parasites The recorded predators of Mycteroperca bonaci include sandbar shark (Carcharhinus plumbeus) and great hamme//...//

Top 3

## Section Title: Utilisation Mycteroperca bonaci is quite tasty and an important food fish. It is fished for sale and for sport.//...//

Top 1

## Section Title: They are considered to be a species of least concern in the United States and in The Bahamas, but considered to be **near threatened** //...//

Top 2

## Section Title: Conservation Mycteroperca bonaci is an IUCN Red List **near threatened** species, vulnerable to increases because it is a relatively slow breeder.

## Section Title: Pacific halibut Hippoglossus stenolepis, the Pacific halibut, is a species of righteye flounder. This very large species of //...//

Top 3

## Section Title: Mycteroperca bonaci Mycteroperca bonaci, the black grouper, black rockfish or marbled rockfish, is a species of marine ray-finned fish//...//

**Concat Top Tool Sections**

**Filtering Stage**

**Tool Sections:** They are considered to be a species of least concern in the United States and in The Bahamas, but considered to be **near threatened** in the western Atlantic Ocean. because of their vulnerable status in South America and reported threats throughout many areas of Central America and the Caribbean. //...//

**Filtering Query:** Here is the user question, What is the conservation status of this fish? \n Here is the relevant information retrieved through image retrieval, Mycteroperca bonaci, the black grouper, black rockfish or marbled rockfish, is a species of marine ray-finned fish, a grouper from the subfamily //...// . Here is the relevant information through Captioning and Grounding.//......[Top Tool Sections]...// \n To obtain useful information, you must conduct reasoning inside first every time you get new retrieved information. After reasoning, you should provide the filtered information inside and , without detailed illustrations.

**Filtering**    **VLM-PRF**

**Filtering Result:** <think>//...//</think> <answer> The conservation status of the black grouper is **Near Threatened**. </answer>

**VLM-Base**

**Wiki-PRF Answer:** The conservation status of the fish in the picture is **Near Threatened**.

**Ground Truth:** ['NT', 'Near Threatened', 'LR/nt']

Figure 12: **Illustration of Wiki-PRF on Question Infoseek_00033513 from Infoseek.**

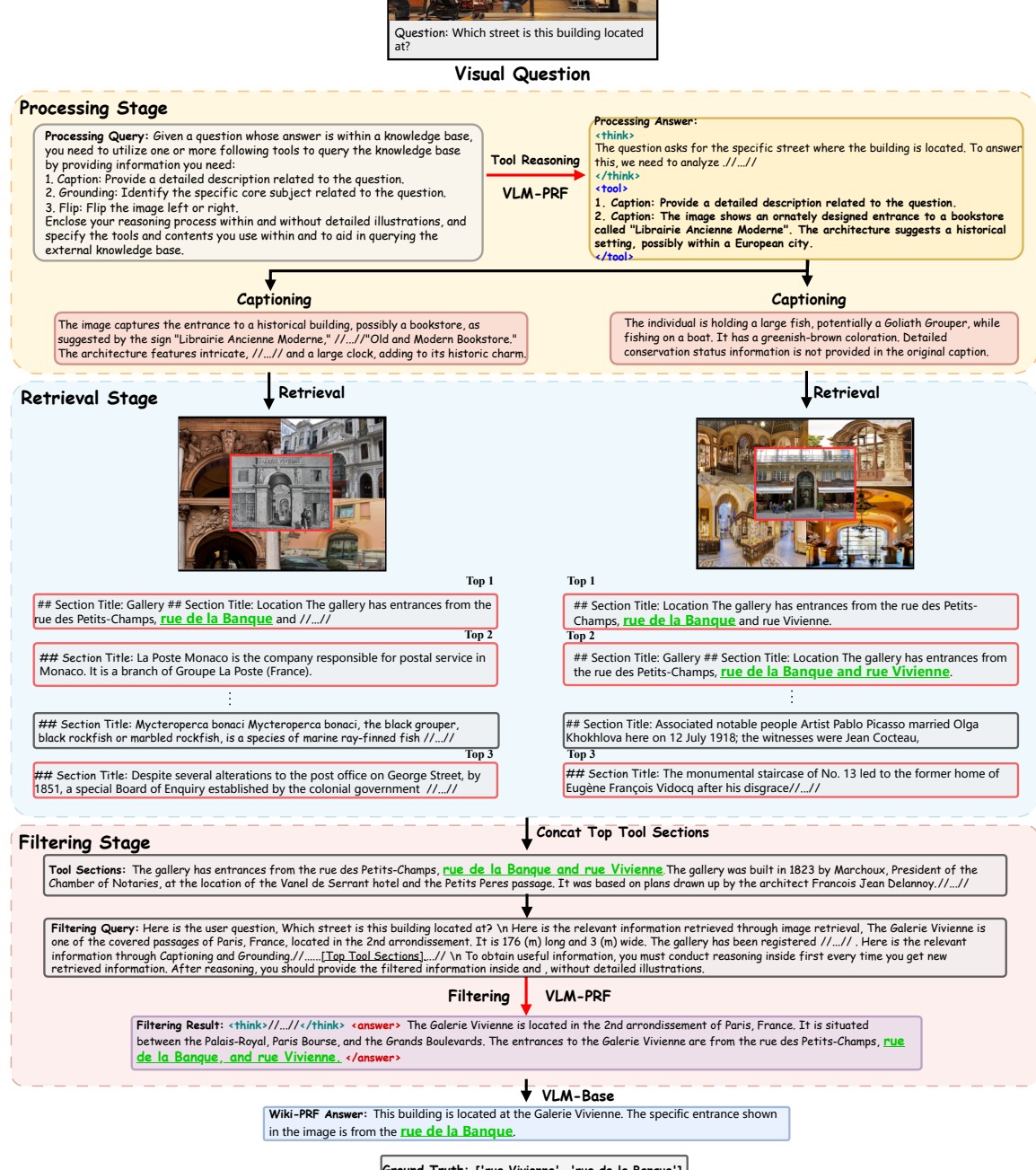

Figure 13: **Illustration of Wiki-PRF on Question Infoseek_00005094 from Infoseek.**

