# OpenReview forum: "Knowledge-based Visual Question Answer with Multimodal Processing, Retrieval and Filtering"
_NeurIPS.cc/2025/Conference — NeurIPS 2025 poster_

### Official Review · Reviewer_CtMV · 2025-06-17

**Clarity:** 4
**Significance:** 2
**Originality:** 3
**Rating:** 3
**Confidence:** 4

**Summary:**

This paper focuses on the task of knowledge-based visual question answering (KB-VQA) and proposes a new three-stage RAG paradigm, which includes multimodal processing, multimodal retrieval, and filtering.

**Questions:**

See the weakness.

**Ethical Concerns:**

["NO or VERY MINOR ethics concerns only"]

**Limitations:**

See the weakness.

**Paper Formatting Concerns:**

No.

**Quality:**

3

**Strengths And Weaknesses:**

Strengths:


The motivation of the paper is clear.
The paper is well-written and easy to follow.


Weaknesses:


The authors claim that KB-VQA involves two distinct challenges. However, based on the discussion in the introduction, these two challenges appear to essentially describe the same underlying issue.

It is unclear why widely-used benchmark datasets for KB-VQA were not selected for evaluation. This omission weakens the experimental validation.

While the paper emphasizes its novel RAG paradigm, the reported improvements seem to primarily stem from reinforcement learning (RL). The proposed RAG approach differs from traditional RAG mainly in the multimodal processing stage (e.g., through image captioning and object detection). It would be helpful for the authors to clarify whether the RL component could be replaced by existing large language models (LLMs). If so, the novelty of this work might be limited.

---

> ### Author Rebuttal · Authors · 2025-07-31
>
> Thank you for your constructive reviews and we will address your concerns as follows.
>
> **Q1:** Differences between the two challenges mentioned in introduction. \
> **A1:** In the introduction, we outlined two current challenges in the Visual RAG problem \[1\]:
>
> 1.  In the retrieval stage, reliance solely on global image-text features prevents fine-grained analysis of the image, resulting in ineffective recall of high quality information.
>
> 2.  In the filtering stage, the retrieved information inevitably contains noise, which current ranking methods cannot fundamentally address, degrading answer quality.
>
>
> These two challenges are addressed by improving the granularity/accuracy of retrieved content and the relevance of retrieval results to the question.
>
> To this end, we designed the Processing and Filtering stages:
>
> 1.  By invoking image processing tools, we enhance fine-grained image understanding for the question and improve accuracy of retrieved content .
>
> 2.  By filtering recalled content using a visual language model, we remove irrelevant/redundant information, thereby improving accuracy of answer .
>
>
> **Q2**: Why widely-used benchmark datasets for KB-VQA were not selected for evaluation?\
> **A2**: In the paper, we evaluated our method on E-VQA and InfoSeek, which are two widely-used benchmarks in KU-RAG, MMKB-RAG, etc. To further validate our method, we experiment on OK-VQA benchmark \[6\] with the standard setup from M2KR \[7\] for a fair comparison.
>
> The results indicate our Wiki-PRF-7B achieves a new state-of-the-art score of 77.8 on OK-VQA.
>
> | Method | Model | OK-VQA | E-VQA | InfoSeek |
> | --- | --- | --- | --- | --- |
> | Qwen2.5-VL-3B |  | 62.1 | 19.6 | 21.4 |
> | Qwen2.5-VL-7B |  | 72.4 | 20.3 | 23.7 |
> | KU-RAG | LLaVA-Next-7B | 73.1 | 11.0 | 9.1 |
> | MMKB-RAG | LLaMA-3.1-8B | 65.4 | 35.9 | 36.4 |
> | **Wiki-PRF-3B** | VLM-PRF-3B | **68.6** | **32.4** | **39.0** |
> | **Wiki-PRF-7B** | VLM-PRF-7B | **77.8** | **36.0** | **42.8** |
>
> **Table R1**: Performance on OK-VQA, E-VQA, and InfoSeek.
>
> **Q3**: Concern about the improvements seem to primarily stem from RL.\
> **A3**: Actually, the performance improvement stems from the combined effects of the novel RAG framework and RL training. To reveal this, we implement a standard RAG based on Qwen2.5-VL as a baseline for evaluation, termed Vanilla-MRAG. As shown in Table R2, when both use the 3B model, our Wiki-PRF-3B (w/o RL) outperforms Vanilla-MRAG-3B by 2.54 points (36.76% vs. 34.22%). After using RL training, an additional 2.72 points of improvement are achieved (39.48% vs. 36.76%). The two improvements have an overall comparable impact on RAG capabilities. Similar results are observed for the 7B model. This demonstrates that our PRF framework plays an important role in improving the overall performance of RAG. We conduct our experiments on a 2K-subset of the InfoSeek test split.
>
> | Method | Model | InfoSeek |
> | --- | --- | --- |
> | Vanilla-MRAG-3B | Qwen2.5-VL-3B | 34.22 |
> | **Wiki-PRF-3B (w/o RL)** | Qwen2.5-VL-3B | **36.76** |
> | **Wiki-PRF-3B (with RL)** | VLM-PRF-3B | **39.48** |
> |Vanilla-MRAG-7B | Qwen2.5-VL-7B | 37.53 |
> | **Wiki-PRF-7B (w/o RL)** | Qwen2.5-VL-7B | **39.71** |
> | **Wiki-PRF-7B (with RL)** | VLM-PRF-7B | **43.12** |
>
> **Table R2:** Ablation of our framework on a 2K-subset of the InfoSeek test split. (w/o: without)
>
> **Q4**: The proposed RAG approach differs from traditional RAG mainly in the multimodal processing stage.\
> **A4**: Actually, the filtering stage is also one of the main differences. We use the filtering stage to replace the reranker module in traditional RAG. As in Tab. 6 and (Ln268-Ln275), our processing stage and filtering stage contribute equally to overall performance. Combining the analysis in A1, results in Table R3 show that the processing stage improves performance by 2.02% over the base model by enhancing recall of highly relevant documents, while the filtering stage achieves another 3.24% gain by filtering the most relevant information. Therefore, in Wiki-PRF, the processing stage and the filtering stage are equally important.
>
> | Processing | Retrieval | Filtering | VQA Accuracy |
> | --- | --- | --- | --- |
> | \- | ✔ | \- | 34.22 |
> | ✔ | ✔ | \- | 36.24 |
> | \- | ✔ | ✔ | 36.76 |
> | ✔ | ✔ | ✔ | **39.48** |
>
> **Table R3:** Ablation of Processing stage and Filtering stage on Wiki-PRF-3B.
>
> **Q5**: Whether the RL component could be replaced by existing LLMs.\
> **A5**: More powerful VLMs will, to some extent, replace the gains from our RL component. However, the RL component is more suitable for our Wiki-PRF framework.  This is because reinforcement learning training can improve model performance without intermediate annotations. Coincidentally, our training data lacks labeled tool combinations (intermediate annotations) due to the high cost of annotation. This creates an implicit and irreplaceable advantage.
>
> Besides, larger VLMs bring greater deployment overhead and API costs in applications. Our RL component, however, can achieve better performance at a lower cost. As shown in Table R4, even Vanilla-MRAG-32B, a model four times larger, cannot achieve the same performance as our Wiki-PRF-7B on InfoSeek.
>
> | Method | Model | InfoSeek (UQ) | InfoSeek (UE) | InfoSeek (All) |
> | --- | --- | --- | --- | --- |
> | Qwen2.5-VL-32B | \- | 33.9 | 32.7 | 32.9 |
> | Vanilla-MRAG-32B [1] | Qwen2.5-VL-32B | 41.2 | 40.3 | 40.4 |
> | Wki-PRF-7B (with RL) | VLM-PRF-7B | **43.3** | **42.7** | **42.8** |
>
> **Table R4**: Performance on larger models.
>
> **References**
>
> \[1\]Lewis, P., Perez, E., Piktus, A., Petroni, F., Karpukhin, V., Goyal, N., ... & Kiela, D. (2020). Retrieval-augmented generation for knowledge-intensive nlp tasks.  (Neurlps 2020)
>
> \[2\] Faysse, M., Sibille, H., Wu, T., Omrani, B., Viaud, G., Hudelot, C., & Colombo, P. (2024). Colpali: Efficient document retrieval with vision language models.  （ICLR2025).
>
> \[3\] Lin, W., Chen, J., Mei, J., Coca, A., & Byrne, B. (2023). Fine-grained late-interaction multi-modal retrieval for retrieval augmented visual question answering.  (Neurlps 2023)
>
> \[4\] Abdallah, A., Piryani, B., Mozafari, J., Ali, M., & Jatowt, A. (2025). Rankify: A comprehensive python toolkit for retrieval, re-ranking, and retrieval-augmented generation. _arXiv preprint arXiv:2502.02464_.
>
> \[5\] Li, N., Liu, J., Shan, Y., Huang, M., & Li, T. (2025). Graph-based RAG Enhancement via Global Query Disambiguation and Dependency-Aware Reranking. _arXiv preprint arXiv:2506.11106_.
>
> \[6\] Marino, K., Rastegari, M., Farhadi, A., & Mottaghi, R. (2019). Ok-vqa: A visual question answering benchmark requiring external knowledge. （CVPR2019）
>
> \[7\] Weizhe Lin, Jingbiao Mei, Jinghong Chen, and Bill Byrne. 2024. PreFLMR: Scaling Up Fine-Grained Late-Interaction Multi-modal Retrievers. (ACL2024)

---

> > ### Comment · Reviewer_CtMV · 2025-08-06
> >
> > Thanks for the detailed response.
> > However, I have some questions about the evaluation setting on the OK-VQA dataset. Since the OK-VQA dataset is the most explored benchmark in the KBVQA task, I think it is essential to clear that out.
> > 1. Since the evaluation follows M2KR in PreFLMR, which is a retrieval benchmark with PRR@K as the main metric [1], is your reported metric PRR@K?
> > [1] https://github.com/LinWeizheDragon/FLMR/blob/main/examples/example_use_preflmr.py
> >
> > 2. I have conducted experiments with Qwen2.5-VL-7B on OK-VQA. The VQA score is 61.6/62.0 (zero-shot/sft). The difference between 62.0 and your reported backbone 72.4 is significant. Similar results with Qwen2-VL-7B are also reported in MMKB-RAG, all under 70.0.
> > It is questionable whether the recall is misreported as the VQA score.
> >
> > 3. Qwen2.5-VL and KU-RAG (not in their original paper) are both implemented by the authors. I think if the recall is misreported, these results are not reliable for demonstrating SOTA performance.
> > To my understanding, in Table R1, the MMKB-RAG (65.4) is the VQA score, and others are all PRR@K. Is it right?
> >
> > 4. The published PRR@K SOTA on OK-VQA is 92.6 [2], which is a lot higher than the reported 77.8, with the same retrieval-augmented generation architectures as this paper.
> > [2] Retrieval-Augmented Visual Question Answering via Built-in Autoregressive Search Engines
> >
> > The evaluation of other datasets in the paper is valid to demonstrate their effectiveness. But the new SOTA on OK-VQA is questionable to me.

---

> ### Author Response · Authors · 2025-08-04
>
> Dear Reviewer CtMV,
>
> Thank you again for your time and effort in reviewing our paper. As the discussion phase is drawing to a close, we would like to kindly ask whether our rebuttal has adequately addressed your concerns. If there are any remaining questions or points that need further clarification, we would be happy to provide additional details. If you feel that your concerns have been resolved, we would sincerely appreciate it if you would consider updating your score. We would also greatly value the opportunity to continue the discussion if needed.

---

> ### Author Response · Authors · 2025-08-07
>
> Dear Reviewer CtMV,
>
> Thank you for your detailed feedback. For clarity, all the metrics we reported on the OK-VQA dataset are **VQA Score**, calculated as `Accuracy = min(number of human agreements / 3, 1)`. This score measures the agreement between predicted answers and ground truth answers, where "number of human agreements" is the number of 10 annotators whose answer exactly matches the model's prediction. To address your points in order:
>
> 1.  No, we report the VQA Score metric following the official evaluation of M2KR repository\[1\].
>
> 2.  We list our inference details as follows: We utilized the prompt `Question: {Question} \n Short Answer:` with a temperature of `0.7`. The model is deployed using `vllm==0.9.0.1` on NVIDIA 8xA100-80G GPUs with the command below:
>
>
> ```bash
> vllm serve ${MODEL_DIR} --port 8000 --served-model-name $MODEL_NAME --tensor-parallel-size 8 --gpu-memory-utilization 0.9 --dtype bfloat16 --max-num-batched-tokens 8192 --max-num-seqs 10 --trust-remote-code
> ```
>
> Notably, the KU-RAG paper \[2\] reports in Table 2 that the VQA Scores of OK-VQA dataset for the base models (LLaVA-Next-7B, Qwen2.5-VL-32B, and GPT-4o) on the OK-VQA dataset are all above 70.
>
> 3.  Regarding the results in Table R1, the baseline for Qwen2.5-VL was reproduced by us, while the KU-RAG result (73.1) is cited from Table 2 of the KU-RAG paper \[2\] (entry: "LLaVA-NeXT-7B + KU-RAG"). All listed values are VQA Scores.
>
> 4. To more directly evaluate the impact of different RAG methods on VQA performance, we use the VQA score instead of PRR@K to assess the final answer accuracy throughout this paper. For the VQA Score, the KU-RAG paper\[2\] (released at 25.07) itself reported the previous state-of-the-art VQA Score on OK-VQA as 66.10 and it demonstrated its SOTA performance as 73.1. In comparison, our Wiki-PRF-7B model achieves a score of **77.8**, setting a new state-of-the-art for models in this parameter (<=8B) class.
>
>
> Overall, we use the VQA Score to evaluate the OK-VQA results, and we will release the complete inference and evaluation code.
>
> Reference:
>
>  \[1\] Weizhe Lin, Jingbiao Mei, Jinghong Chen, and Bill Byrne. 2024. PreFLMR: Scaling Up Fine-Grained Late-Interaction Multi-modal Retrievers. (ACL2024)
>
>  \[2\] Zhang, Z., Wu, Y., Luo, Y., & Tang, N. (2025). Fine-Grained Knowledge Structuring and Retrieval for Visual Question Answering.

---

> ### Author Response · Authors · 2025-08-09
> **Acknowledgments**
>
> Dear Reviewer CtMV,
>
> As our discussion nears its conclusion, we would like to sincerely thank you for your thoughtful comments and discussion. If your concerns have been addressed, we would be truly grateful if you could kindly consider raising the score. If you have any further questions, we are pleased to address your concerns.

---

### Official Review · Reviewer_VEoU · 2025-07-02

**Clarity:** 3
**Significance:** 3
**Originality:** 3
**Rating:** 4
**Confidence:** 4

**Summary:**

This paper introduces a three-stage framework for knowledge-based visual question answering. The first stage processes the raw query with several tools, including captioning, grounding, and flipping to obtain revised queries. Then it uses multi-modal retrieval to obtain an initial set of retrieved documents with the revised queries. A VLM-based filtering is used to summarize and generate informative knowledge representation from the initial document set, and the selected documents are used for final answer generation. The filtering is trained from reinforcement learning using the reward designed from the final answer generation. The authors conduct experiments on E-VQA and InfoSeek to illustrate the effectiveness of the proposed method.

**Questions:**

1. Is the VLM used for final answer generation a fixed VLM model?

2. From the inference time, it seems that the filtering stage is the most time-consuming part. Is it because of the long input or the reasoning part?

3. Can you provide the training curve so that we can have an idea regarding the stability of the RL training?

**Ethical Concerns:**

["NO or VERY MINOR ethics concerns only"]

**Final Justification:**

I decided to keep my positive score after reading the authors' response and other reviewers' comments. The author have clarified my concerns with additional experiments. I also think most other reviewers' comments have been addressed.

**Limitations:**

Yes

**Quality:**

3

**Strengths And Weaknesses:**

**Strengths**

1. It is an interesting idea to use different tools for query expansion and use RL to guide the selection of useful knowledge representation. Those directions may provide new insights into the multi-modal retrieval-augmented generation framework design.

2. The authors conduct extensive experiments on E-VQA and InfoSeek to compare with some strong models. They also provide a detailed analysis of the retrieval performance, filtering performance, and ablation study for different tools.

**Weaknesses**

1. One weakness is the verification of the effects of RL. I appreciate that the authors provide Table 5 to verify the effectiveness of the filtering. But since you have a set of "ground-truth" entities, it seems to me that a more straightforward way would be to train the answer filtering to provide context regarding the ground-truth entities, rather than performing complex RL-based training. I would expect the authors to provide the comparison or provide additional explanations that it is not feasible. It would help us better understand the rationale for using the RL-based training.

2. There are a couple of narratives that require additional explanations:
- The authors choose GRPO with the removed KL divergence. Why do you remove the KL penalty?
- The design of the reward also includes multiple terms. It would be beneficial if the authors could conduct an ablation study on the reward design.
- The authors use another VLM for final answer generation. Why do you use two passes rather than complete the knowledge filtering and answer generation in one LLM?

---

> ### Author Rebuttal · Authors · 2025-07-31
>
> Thank you for your constructive reviews and we will address your concerns as follows.
>
> **Q1**: Verification of the effects of RL\
> **A1**: Thank you for pointing this out. To investigate its effectiveness, we evaluated using SFT for the filtering stage. We trained a dedicated filtering model with SFT, maintaining the same configuration as our RL setup. The results are as in Table R1.
>
> The RL model significantly outperforms the SFT model on the test set. We believe this is because SFT tends to imitate surface-level patterns, limiting its generalization capabilities. In contrast, RL enables the model to grasp the underlying principles of information filtering, leading to a far more robust and generalizable performance.
>
>
> | Model | Unseen Question (UQ) | Unseen Entity (UE) | ALL |
> | --- | --- | --- | --- |
> | Qwen2.5-VL-7B | 39.1 | 40.5 | 40.2 |
> | Wiki-PRF-7B (SFT) | 41.5 | 41.9 | 41.8 |
> | **Wiki-PRF-7B (RL)** | **46.6** | **46.2** | **46.3** |
>
> **Table R1**: Performance on the Test set.
>
> **Q2**: Why do you remove the KL penalty?\
> **A2**:
> 1.  To maximize exploration of tool usage, we remove the KL divergence term, which constraints the output distribution of the trained model to remain close to the base model while inherently limiting exploration. We explore tool-usage combinations while supervising result correctness via answer rewards during training.
>
> 2.  Following DAPO \[1\], we adopt KL divergence removal for unconstrained exploration, which is a strategy validated by recent work \[2–4\] that enhances exploration through this approach.
>
>
> **Q3**: The design of the reward also includes multiple terms. It would be beneficial if the authors could conduct an ablation study on the reward design.\
> **A3**: The reward weight ablation study is presented in Appendix C.3 (L68-72). We show the ablation result in the table below. Results demonstrate that a 1:1 weighting of reward terms yields optimal performance. This confirms both format and answer rewards are essential.
>
> | Method|3:1 | 2:1 | 1:1 | 1:2 | 1:3 |
> | --- | --- | --- | --- | --- | --- |
> | Wiki-PRF-3B | 38.80 | 38.20 | 39.48 | 38.89 | 38.53 |
>
> **Table R2:** Performance on different reward designs. X:Y means the weight ratio of answer reward and format reward.
>
> **Q4**: Why do you use two passes rather than complete the knowledge filtering and answer generation in one LLM?\
> **A4**: After retrieval, our method retrieves substantial information, including both relevant content and semantically similar but irrelevant noise. Inspired by long-text processing techniques, we propose inserting a VLM-based filtering stage before answer generation. This filter selects relevant passages, removes noise, and then generates answers using the filtered information. Experiment results in Table R3 demonstrates that our method significantly improves question-answering accuracy.
>
> | Method | Processing | Filtering | VQA Accuracy |
> | --- | --- | --- | --- |
> | Wiki-PRF-3B | ✔ | \- | 36.76 |
> | Wiki-PRF-3B | ✔ | ✔ | 39.48 |
>
> **Table R3** Ablation of Filtering
>
> **Q5**: Is the VLM used for final answer generation a fixed VLM model?\
> **A5**: The VLM used for final answer generation is flexible. But our framework employs a single vision-language backbone across all stages, enabling parameter sharing that reduces total parameters by half while maintaining inference efficiency through weight inheritance. This design eliminates redundant model instantiation, directly addressing scalability constraints in multimodal RAG systems.
>
> **Q6**: Why is the filtering stage is the most time-consuming part?\
> **A6**: As in Table 9 (L293-L294), compared to the reasoning part, processing a large amount of retrieval information (i.e., long input) takes more time. This makes the filtering stage the most time-consuming part.
>
> **Q7**: Can you provide the training curve so that we can have an idea regarding the stability of the RL training?\
> **A7**: We provide the training curves in Appendix C.1 (L53-60) demonstraing the convergence of our RL training process.  As shown in the figure, the answer reward and format reward exhibit stable improvement, while the number of tokens decreases steadily over training. It indicates that the RL successfully optimizes both answer quality and conciseness.
>
> **References**
>
> \[1\] Yu, Q., Zhang, Z., Zhu, R., Yuan, Y., Zuo, X., Yue, Y., ... & Wang, M. (2025). Dapo: An open-source llm reinforcement learning system at scale. _arXiv preprint arXiv:2503.14476_. (2025).
>
> \[2\] Zhang, X., Sun, H., Zhang, Y., Feng, K., Lu, C., Yang, C., & Meng, H. (2025). Critique-GRPO: Advancing LLM Reasoning with Natural Language and Numerical Feedback. _arXiv preprint arXiv:2506.03106_.
>
> \[3\] Chu, X., Huang, H., Zhang, X., Wei, F., & Wang, Y. (2025). Gpg: A simple and strong reinforcement learning baseline for model reasoning. _arXiv preprint arXiv:2504.02546_.
>
> \[4\] Liu, Z., Chen, C., Li, W., Qi, P., Pang, T., Du, C., ... & Lin, M. (2025). Understanding r1-zero-like training: A critical perspective. _arXiv preprint arXiv:2503.20783_.

---

> > ### Comment · Reviewer_VEoU · 2025-08-05
> > **Response to authors**
> >
> > Thanks for the response. My concerns have been addressed.

---

> ### Author Response · Authors · 2025-08-04
>
> Dear Reviewer VEoU,
>
> Thank you again for your time and effort in reviewing our paper. As the discussion phase is nearing its end, we kindly ask whether our rebuttal has sufficiently addressed your concerns. If there are any remaining questions or points requiring clarification, we would be happy to provide further details. We would greatly appreciate the opportunity to engage in further discussion with you.

---

> ### Author Response · Authors · 2025-08-09
> **Acknowledgments**
>
> Dear Reviewer VEoU,
>
> Thank you once again for your valuable feedback and the thoughtful discussion. We truly appreciate your insights.

---

### Official Review · Reviewer_bmZr · 2025-07-02

**Clarity:** 3
**Significance:** 2
**Originality:** 3
**Rating:** 4
**Confidence:** 4

**Summary:**

This paper proposes Wiki-PRF, a three-stage framework (Processing, Retrieval, Filtering) for knowledge-based visual question answering (KB-VQA). By dynamically invoking visual tools (e.g., grounding, captioning) to extract precise multimodal features, integrating visual-textual retrieval, and applying reinforcement learning (RL) for relevance filtering, the method addresses challenges in retrieval-augmented generation (RAG), such as imprecise queries and noisy results. Experiments on E-VQA and InfoSeek datasets show that Wiki-PRF achieves state-of-the-art accuracy (36.0 and 42.8, respectively), outperforming prior RAG and vision-language models. The RL-trained VLM-PRF enhances reasoning and tool adaptation, demonstrating robust performance with minimal training data.

**Questions:**

Besides the weaknesses mentioned above, there are still some questions that are expected to be addressed in the rebuttal.
1. In Figure 2, what is the specific VLM model used in the Filtering Stage? It seems that its inputs are pure text. Why not use an LLM instead?
2. Could you provide comparisons with recent multimodal LLMs (e.g., GPT-4O, InternVL3) and discuss compatibility with the latest commercial VLM?
3. Prior works like MMKB-RAG have conducted experiments on the OKVQA and M2KR datasets. I suggest adding results on these two commonly used datasets.

**Ethical Concerns:**

["NO or VERY MINOR ethics concerns only"]

**Final Justification:**

The meticulous response from the authors has addressed my concerns, and I believe the proposed multi-stage framework is reasonable. On the other hand, the relatively complex pipeline hinders the practicality and reproducibility. Therefore, I would maintain my initial rating as borderline accept.

**Limitations:**

yes

**Quality:**

3

**Strengths And Weaknesses:**

Strengths:
1. The three-stage design (Processing-Retrieval-Filtering) innovatively combines multimodal tool processing with RL-based filtering, addressing key limitations in traditional RAG for KB-VQA.
2. The RL-based training improves the model's reasoning ability without extensive labeled data.
3. Comprehensive experiments and ablation studies validate the effectiveness of each stage and tool, with clear performance gains over state-of-the-art baselines (e.g., ReflectiVA, MMKB-RAG).

Weaknesses:
1. The framework relies on a limited set of tools (captioning, grounding, flipping), which may not cover all KB-VQA scenarios.
2. The comparisons in Table 2 are not fair, as the base LLMs are different. For rigor, I suggest using the same LLaMA-3.1-8B model to compare with ReflectiVA and MMKB-RAG.
3. The terms Wiki-PRF and VLM-PRF are confusing. I don't think it is necessary to use two different names.

---

> ### Author Rebuttal · Authors · 2025-07-31
>
> Thank you for your constructive reviews and we will address your concerns as follows.
>
> **Q1**: The toolset used within the framework is limited.\
> **A1**: In this study, we have chosen  three highly relevant tools (flipping, captioning, and grounding) for KB-VQA. However, our framework is flexible, not confined to a specific toolset, but rather tailored to the specific needs of the task. We will explore more tools for other tasks in future work.
>
> **Q2**: Use the same base LLMs (i.e., LLaMA-3.1-8B) for comparision. Provide comparisons with recent multimodal LLMs (e.g., GPT-4O, InternVL3) and discuss compatibility with the latest commercial VLMs.\
> **A2**：Thank you for pointing this out.
>
> 1.  We supplement a new set of experiments using LLaMA-3.1-8B as a unified backbone to fairly compare our Wiki-PRF framework with other methods.
>
> 2.  We extend this analysis to an additional powerful base model (i.e., InternVL3-8B) and compare against the latest commercial VLMs (i.e., GPT-4o) to demonstrate our method's general applicability.
>
> The table  R1 demonstrates that under identical LLaMA-3.1-8B base model, Wiki-PRF achieves superior performance with consistent gains over ReflectiVA (+8.3 on EVQA, +0.7 on InfoSeek) and MMKB-RAG (+4.4 on InfoSeek), validating its efficacy.
>
> Moreover, Wiki-PRF adapts to different base models and can achieve consistent improvements. With a stronger base model (InternVL3-8B), our framework achieves a new SOTA score of 39.2% on E-VQA, surpassing all competitors. We will incorporate these comparisons in the Table 2.
>
> The updated results are presented below:
>
> | Method | Model | E-VQA (Single) | E-VQA (All) | InfoSeek (UQ) | InfoSeek (UE) | InfoSeek (All) |
> | --- | --- | --- | --- | --- | --- | --- |
> | GPT-4o | \- | \- | 15.2 | \- | \- | 36.1 |
> | LLaMA-3.1-8B | \- | 20.3 | 20.4 | 8.8 | 7.6 | 7.9 |
> | Qwen2.5-VL-7B | \- | 21.7 | 20.3 | 22.8 | 24.1 | 23.7 |
> | InternVL3-8B | \- | 24.3 | 23.7 | 24.2 | 21.4 | 22.0 |
> | ReflectiVA | LLaMA-3.1-8B | 28.0 | 29.2 | 40.4 | 39.8 | 40.1 |
> | MMKB-RAG | LLaMA-3.1-8B | 39.7 | 35.9 | 36.4 | 36.3 | 36.4 |
> | Wiki-PRF | LLaMA-3.1-8B | 36.3 | 35.5 | 41.3 | 40.6 | 40.8 |
> | Wiki-PRF | Qwen2.5-VL-7B | 37.1 | 36.0 | 43.3 | **42.7** | **42.8** |
> | Wiki-PRF | InternVL3-8B | **40.1** | **39.2** | **43.5** | 42.1 | 42.5 |
>
> **Table R1:** Performance on different base models.
>
> **Q3**: The terms Wiki-PRF and VLM-PRF are confusing.\
> **A3**: We use Wiki-PRF and VLM-PRF to distinguish the method from the model.
>
> *   Wiki-PRF is our proposed framework (the full Processing-Retrieval-Filtering pipeline), as in L47.
>
> *   VLM-PRF is the specific model that is trained via RL for tool-use and information-filtering within this framework, as in L52.
>
>
> **Q4**: What is the specific VLM model used in the Filtering Stage? It seems that its inputs are pure text. Why not use an LLM instead?\
> **A4**:
>
> 1.  The VLM model used in the Filtering Stage is our VLM-PRF trained on Qwen2.5-VL-7B, same as the model in the Processing Stage.
>
> 2.  The filtering stage requires the image as a critical verification source to align visual content with retrieved information. For example, when answering "What species is the green bird in the image?" The visual grounding constitutes a fundamental requirement that pure LLMs cannot fulfill. Therefore, the VLM cannot be replaced with LLM. We will highlight the input image in Fig. 4.
>
>
> **Q5**: Suggesting more experiments on the OKVQA and M2KR datasets.\
> **A5**: Thank you for the suggestion. Table R2 presents the evaluation of Wiki-PRF on OK-VQA benchmark. It is worth noting that M2KR\[2\] is an evaluation framework for OK-VQA.
>
> Our method demonstrates consistent generalization, with our Wiki-PRF trained on Qwen2.5-VL-7B achieving a new state-of-the-art result of 77.8% (+4.7% and +12.4% over KU-RAG and MMKB-RAG, respectively).
>
> | Method | Model | OK-VQA | E-VQA | InfoSeek |
> | --- | --- | --- | --- | --- |
> | Qwen2.5-VL-3B |  | 62.1 | 19.6 | 21.4 |
> | Qwen2.5-VL-7B |  | 72.4 | 20.3 | 23.7 |
> | KU-RAG | LLaVA-Next-7B | 73.1 | 11.0 | 9.1 |
> | MMKB-RAG | LLaMA-3.1-8B | 65.4 | 35.9 | 36.4 |
> | **Wiki-PRF-3B** | **VLM-PRF-3B** | **68.6** | **32.4** | **39.0** |
> | **Wiki-PRF-7B** | **VLM-PRF-7B** | **77.8** | **36.0** | **42.8** |
>
> **Table R2:** Performance on OK-VQA, E-VQA, and InfoSeek.
>
>
> **References**
>
> \[1\] Marino, K., Rastegari, M., Farhadi, A., & Mottaghi, R. (2019). Ok-vqa: A visual question answering benchmark requiring external knowledge. （CVPR2019）
>
> \[2\] Weizhe Lin, Jingbiao Mei, Jinghong Chen, and Bill Byrne. 2024. PreFLMR: Scaling Up Fine-Grained Late-Interaction Multi-modal Retrievers. (ACL2024)

---

> > ### Author Response · Authors · 2025-08-04
> >
> > Dear Reviewer bmZr,
> >
> > Thank you again for your time and effort in reviewing our paper. As the discussion phase is nearing its end, we kindly ask whether our rebuttal has sufficiently addressed your concerns. If there are any remaining questions or points requiring clarification, we would be happy to provide further details. We would greatly appreciate the opportunity to engage in further discussion with you.

---

> > ### Comment · Reviewer_bmZr · 2025-08-04
> >
> > I appreciate the efforts made by the authors. Most of my concerns are resolved during the response.
> >
> > About the M2KR benchmark, it is worth noting that M2KR is a multimodal knowledge retrieval benchmark, and other related works mentioned in the paper (such as MMKB-RAG) all use this benchmark. I suggest adding experiments on this benchmark to show advantages over previous works.

---

> ### Author Response · Authors · 2025-08-05
>
> Dear Reviewer bmZr,
>
> Thank you for your appreciate and suggestion.
>
> *   Following your suggestion, we carefully re-examine M2KR \[1\]. As stated in _**Section 3**_ \[1\]: _"To properly study general-purpose multi-modal retrievers, we introduce the Multi-task Multi-modal Knowledge Retrieval (M2KR) benchmark suite. We convert nine diverse datasets, originally designed for vision and language tasks such as image recognition, image captioning, and conversational interactions, into a uniform retrieval format."_ This is further corroborated by _Table 2_ in M2KR \[1\].  Therefore, M2KR is a benchmark suite that consolidates multiple existing KB-VQA datasets, rather than being a completely new benchmark.
>
>
> *   As we focus on  image+text to text retrieval tasks, we consulted the official M2KR website, where _Challenge Task 2_ _(Image+Text)_ explicitly includes the datasets we used: OK-VQA, Infoseek, and E-VQA. The dataset splits are clearly listed in the files provided in the official Hugging Face repository.  Following your valuable suggestion, we have expanded our evaluation to include OK-VQA in the rebuttal (Table R2), following the standard M2KR protocols and metrics. We note that, based on average metrics, InfoSeek and E-VQA are more challenging, which is why they have been the primary focus of recent related works \[2,3,4\].
>
> *   Regarding the use of M2KR datasets in **MMKB-RAG**\[2\], we kindly refer you to _**Section**_ _**4.5 Comparison on M2KR dataset\[2\]**_ and _**Table 6**\[2\]_, which adopts the same data processing and evaluation methodology as our work for image+text to-text retrieval tasks in M2KR. and reformatted them as Table 6 for consistency with MMKB-RGA. However, we have noticed discrepancies in the evaluation metrics of M2KR used for E-VQA compared to the standard, and we are currently reviewing the relevant code and data to resolve this issue. The results demonstrate that our method surpasses MMKB-RAG and achieves state-of-the-art performance on OK-VQA.
>
> We will incorporate OK-VQA results and the description of M2KR benchmark in our final manuscript. We  appreciate your thoughtful feedback. Your support and constructive discussion are invaluable to the improvement  of our work.
>
> **Table 6:** Performance comparison on the M2KR dataset.
>
> | Method  | OK-VQA | Infoseek |
> | --- | --- |--- |
> |**zero-shot MLLMs**|||
> |RA-VQAv2|55.44|21.78|
> |Qwen2-VL-Instruct|60.45|21.75|
> |Qwen2-VL-Instruct(sft)|64.08|26.00|
> |||||
> |**Retrieval-Augmented Models**|||
> |RA-VQAv2 w/ FLMR|60.75|-|
> |RA-VQAv2 w/ PreFLMR|61.88|30.65|
> |Qwen2-VL-Instruct w/ PreFLMR|46.99|24.68|
> |Qwen2-VL-Instruct(sft) w/ PreFLMR|65.07|30.74|
> |MMKB-RAG w/ PreFLMR|65.44|34.72|
> | **Wiki-PRF-3B**  | **68.6** | **39.0** |
> | **Wiki-PRF-7B**   | **77.8** | **42.8** |
>
> \[1\] Weizhe Lin, Jingbiao Mei, Jinghong Chen, and Bill Byrne. 2024. PreFLMR: Scaling Up Fine-Grained Late-Interaction Multi-modal Retrievers. (ACL2024)
>
> \[2\] Ling, Zihan, et al. "MMKB-RAG: A Multi-Modal Knowledge-Based Retrieval-Augmented Generation Framework." (arXiv 2025).
>
> \[3\] Yan, Yibin, and Weidi Xie. "EchoSight: Advancing visual-language models with Wiki knowledge." (EMNLP 2024).
>
> \[4\] Cocchi, Federico, et al. "Augmenting multimodal llms with self-reflective tokens for knowledge-based visual question answering." (_CVPR_ 2025.)

---

> ### Author Response · Authors · 2025-08-09
> **Acknowledgments**
>
> Dear Reviewer bmZr,
>
> Thank you once again for your valuable feedback and the thoughtful discussion. We truly appreciate your insights.

---

### Official Review · Reviewer_aWe7 · 2025-07-02

**Clarity:** 2
**Significance:** 3
**Originality:** 2
**Rating:** 4
**Confidence:** 3

**Summary:**

This paper presents a retrieval-augmented generation (RAG) method named Wiki-PRF to address knowledge-based visual question answering (KB-VQA). To improve multimodal query quality and the relevance of retrieved results, Wiki-PRF utilizes a three-stage method consisting of Processing, Retrieval, and Filtering stages.

In the Processing stage, a model, named VLM-PRF, invokes visual tools (captioning, grounding, flipping) to convert original images and questions into retrieval queries. The Retrieval stage uses EVA-CLIP and FAISS to perform multimodal knowledge retrieval on Wikipedia articles. The Filtering stage, VLM-PRF is used to filter irrelevant information and retain the most relevant content

The training of VLM-PRF is basically borrowed from the DeepSeek-R1 and its GRPO algorithm.
Experiments on E-VQA and InfoSeek datasets show significant improvements, achieving state-of-the-art performance. Ablation studies validate the effectiveness of each stage and tool, while analyses demonstrate the efficiency of RL training with limited samples.

**Questions:**

Major concerns are raised in the Weaknesses section. Besides, I would like to ask the following questions:

1. Why trains a single model to perform two tasks (query generation based on tool-use and retrieved results filtering) via RL? What if we train two separate models, one for each task?
2. What is the scale of the knowledge base used for retrieval? How does the performance change with larger scale knowledge bases?
3. In Table 4, what does "combinations" mean? why the sum of mean usage of all three tools is not approximately equal to the combination value?

**Ethical Concerns:**

["NO or VERY MINOR ethics concerns only"]

**Final Justification:**

Thank you to the Authors for your patient rebuttal; my concerns have now largely been addressed. The experimental results and conclusions (1. MM RAG can be improved by RLVR; 2. MM RAG is helpful for knowledge-based VQA) of the paper are convincing. Thus, I believe the paper is worthy of acceptance.

**Limitations:**

yes

**Quality:**

2

**Strengths And Weaknesses:**

**Strengths**:
- Wiki-PRF’s three-stage pipeline, Processing-Retrieval-Filtering, is succinct and effective to address the challenges of multimodal RAG in KB-VQA. Leveraging visual cues to boosts retrieval quality, through tool-based query generation (e.g., grounding to focus on relevant regions) and VLM-powered result filtering.
- As far as I can tell, in the field of KB-VQA, this is the first attempt to build a multimodal RAG workflow that is enhanced by reinforcement learning and tool-use techniques.
- Experiments on benchmark datasets (E-VQA, InfoSeek) demonstrate consistent SOTA results. Ablation studies confirm the contributions of each stage and tools.

**Weaknesses**:
- **Limited novelty**: Despite the decent performance, the approach lacks novelty. It mainly combines existing techniques (RAG, RL, tool-use) without introducing significant new concepts or methodologies, thus it is valuable in terms of engineering practice but lacks academic contribution.
- **Unclear description for Processing stage**: Section 3.2 does not clearly explain how the Processing stage works. For example, it is confusing how VLM is involved multiple times (the invocation order and input-output relationships). $Caption_{init}$ is also not defined.
- **Missing details of training data**: The paper does not provide sufficient details on the training data used for VLM-PRF, which are crucial since RL training requires labeled data. It is unclear how the training data is constructed to contain the ground-truth of tool-use and filtering.
- **Lack of Generalization Analysis**: While effective on E-VQA and InfoSeek, the paper does not thoroughly explore performance on other KB-VQA datasets or more diverse visual domains (e.g., medical images, satellite imagery).

---

> ### Author Rebuttal · Authors · 2025-07-31
>
> Thank you for your constructive reviews and we will address your concerns as follows.
>
> **Q1**: Efficient engineering practice with limited academic contribution.\
> **A1**: In this work, we recognize two fundamental challenges in KB-VQA problem: (1) fine-grained knowledge retrieval in complex visual scenes and (2) precise filtering of irrelevant information from large scale retrieved results. Based on the challenges above, our framework features three core innovations:
>
> 1.  For coarse retrieval limitations, we innovatively explored a tool-based fine-grained retrieval mechanism.
>
> 2.  For irrelevant information, we innovatively employ a question-based filtering stage, rather than simple reranking\[5\].
>
> 3.  For the lack of extensive labeled data, we innovatively employ RL to train the above stages.
>
>
> The Processing-Retrieval-Filtering (PRF) framework resolves critical KB-VQA limitations in RAG, not only through architectural design but also via component-level technical innovations, enhances multi-step reasoning capability without extensive labeled data (Reviewer bmZr), and provides foundational insights for multimodal RAG design (Reviewer VEoU).
>
> **Q2**: More detailed description for Processing stage.\
> **A2**:  We appreciate the reviewer’s feedback. The function of the VLM-PRF in the Processing stage is to generate an execution tool-usage plan based on the visual and textual inputs. As referenced in L131-133, given the image and question, the VLM-PRF outputs selected tools and execution order in `<tool>` tags.
>
> ---
>
> *   Input: {Image, Question}
>
> *   Output: {`<think>`Thought Process`</think>` `<tool>` 1. ToolA: ContextA; 2. ToolB: ContextB; ...  `</tool>` }
>
>
> ---
>
> Comprehensive case study of InfoSeek and E-VQA illustrating this mechanism is available in the page 7-12 of supplementary material. We will add more detailed description in L131-133 to help understand the work of the processing stage.
>
> **Q3**: How VLM is involved multiple times.\
> **A3**: After VLM-PRF plans the sequence of tool calls, the tasks are executed by VLM-base, a foundational model (Qwen2.5-VL-7B). This model is invoked multiple times to power specific tools like captioning and grounding. In essence, VLM-PRF provides the strategy while VLM-base delivers the core tool functionality, which explains its multiple involvements in the processing stage. We will further clarify the distinct roles of VLM-PRF and VLM-base in Section 3.2 in the revision.
>
> **Q4**: The definition of $Caption\_{init}$.\
> **A4**: Thank you for pointing this out. $Caption\_{init}$ is the specific instruction for the caption tool when VLM-PRF calls it. For example, in a complex photo of flowers, it might be 'Please describe the small insect on the purple flower'. The VLM-Base then executes the caption tool, taking this instruction as input to produce the final caption. This allows our system to focus on relevant details. We will add this definition in L135 and annotate it detailed in Fig. 3.
>
> **Q5**: Missing details of training data.\
> **A5**: Thank you for pointing this out. The training data follows the basic VQA format in EVQ-A \[1\] and Infoseek \[2\].
>
> *   Input: {Image, Question}
>
> *   Label: {Answer}
>
>
> We clarify that our Wiki-PRF model requires no ground-truth labels for tool use or information filtering, as it learns these skills via RL. The model is guided by a reward signal derived solely from the final answer's correctness, which incentivizes it to autonomously develop effective strategies for tool invocation and content filtering. We will add details about the training data at L184.
>
> **Q6**: Generalization Analysis on other KB-VQA datasets.\
> **A6**: To address the concern about generalization, we evaluate our model on the widely-used OK-VQA benchmark \[3\], following the standard setup from M2KR \[4\] for a fair comparison.
>
> We can see that our Wiki-PRF-7B achieves a new state-of-the-art score of 77.8 on OK-VQA. The consistent performance improvement on multiple benchmarks confirms our method's strong generalization ability.
>
> Due to limited time and lack of suitable readily available benchmarks, we leave the evaluation of KB-VQA on medical/satellite images to future work. However, based on our current tests, we believe our approach is still applicable.
>
> | Method | Model | OK-VQA | E-VQA | InfoSeek |
> | --- | --- | --- | --- | --- |
> | Qwen2.5-VL-3B | - | 62.1 | 19.6 | 21.4 |
> | Qwen2.5-VL-7B | - | 72.4 | 20.3 | 23.7 |
> | KU-RAG | LLaVA-Next-7B | 73.1 | 11.0 | 9.1 |
> | MMKB-RAG | LLaMA-3.1-8B | 65.4 | 35.9 | 36.4 |
> | **Wiki-PRF-3B** | VLM-PRF-3B | **68.6** | **32.4** | **39.0** |
> | **Wiki-PRF-7B** | VLM-PRF-7B | **77.8** | **36.0** | **42.8** |
>
> **Table R1:** Performance on OK-VQA, E-VQA, and InfoSeek.
>
> **Q7**: Why trains a single model to perform two tasks via RL? What if we train two separate models, one for each task?\
> **A7**: Our primary goal is to develop a single model for RAG based multimodal QA. This model must not only achieves state-of-the-art accuracy but also considers resource consumption and performance in applications.
>
> 1.  Through task analysis, we implemented joint training via RL to unify image tool selection and retrieved information filtering. Joint training enables the model to generate inputs explicitly optimized for downstream task performance.
>
> 2.  Training two separate models requires deploying two 7B models (14B in total) and doubles training time to ~30 hours. It is a computationally inefficient approach compared to our unified framework.
>
>
> **Q8**: What is the scale of the knowledge base used for retrieval?\
> **A8**: We adopt the identical knowledge base configuration established in \[5\], utilizing 100K Wikipedia articles for InfoSeek and 2M Wikipedia articles for E-VQA.
>
> **Q9**: How does the performance change with larger scale knowledge bases?\
> **A9**: We evaluated our Wiki-PRF on the largest open-source benchmarks, InfoSeek (100k) and E-VQA (2M). Table 2 demonstrates its SOTA performance. To evaluate scalability with limited rebuttal time, we sample varying knowledge base proportions from InfoSeek (10K–100K articles) and evaluate the scalability of 3B and 7B model.
>
> Table R2 below shows that both our method and the baseline exhibit performance degradation as knowledge base size increases. This occurs because larger knowledge bases introduce additional noise, increasing retrieval difficulty, which is a universal challenge for current RAG methods. Critically, Wiki-PRF demonstrates significantly slower degradation for both 3B and 7B models compared to the baseline. Specifically, when scaling from 10K to 100K articles:
>
> *   Wiki-PRF-7B declines by 17.5 percentage points (60.3% -> 42.8%)
>
> *   Vanilla-MRAG (7B) declines by 32.6 percentage points (56.3% -> 23.7%)
>
> *   At 100K articles, Wiki-PRF-7B achieves 42.8% accuracy(1.8× the baseline's 23.7%), demonstrating superior scalability.
>
> | Method | Model | 10K | 50K | 100K |
> | --- | --- | --- | --- | --- |
> | Vanilla-MRAG [6] | Qwen2.5-VL-3B | 49.7% | 32.1% | 21.4% |
> | Wiki-PRF-3B | VLM-PRF-3B | 53.0% | 43.7% | 39.0% |
> | Vanilla-MRAG | Qwen2.5-VL-7B | 56.3% | 39.6% | 23.7% |
> | Wiki-PRF-7B | VLM-PRF-7B | 60.3% | 51.2% | 42.8% |
>
> **Table R2:** Ablation studies on the scale of knowledge base on InfoSeek.
>
> **Q10**: What does "combinations" mean?\
> **A10**: A "combination" refers to aa ordered set of called tools, for example, {grounding, flip, captioning}, {captioning, grounding}, and {grounding, captioning} are three distinct combinations.  The number of tools being combined indicates whether the model has further explored more ways to use the tools through training. As shown in Table 5 in original paper, the model does use a wider variety of ordered tool combinations. We will add the above explanation  to L256.
>
> **Q11**: Why the sum of mean usage of all three tools is not approximately equal to the combination value?\
> **A11**: Because "mean" (per-tool frequency) and "combinations" (distinct sequence count) measure fundamentally different dimensions. The "mean" value per tool represents the average number of times it is invoked per problem instance. For example, a value exceeding 1.0 indicates the model can complete complex tasks through repeated tool invocation. We will add the description of "mean" in L258.
>
> **References**
>
> \[1\] Mensink, T., Uijlings, J., Castrejon, L., Goel, A., Cadar, F., Zhou, H., ... & Ferrari, V. (2023). Encyclopedic vqa: Visual questions about detailed properties of fine-grained categories. (ICCV2023)
>
> \[2\] Chen, Y., Hu, H., Luan, Y., Sun, H., Changpinyo, S., Ritter, A., & Chang, M. W. (2023, January). Can Pre-trained Vision and Language Models Answer Visual Information-Seeking Questions?. (ACL2023)
>
> \[3\] Marino, K., Rastegari, M., Farhadi, A., & Mottaghi, R. (2019). Ok-vqa: A visual question answering benchmark requiring external knowledge. （CVPR2019）.
>
> \[4\] Weizhe Lin, Jingbiao Mei, Jinghong Chen, and Bill Byrne. 2024. PreFLMR: Scaling Up Fine-Grained Late-Interaction Multi-modal Retrievers. (ACL2024)
>
> \[5\] Yan, Yibin, and Weidi Xie. "EchoSight: Advancing visual-language models with Wiki knowledge." (EMNLP 2024).
>
>
> \[6\] Lewis, P., Perez, E., Piktus, A., Petroni, F., Karpukhin, V., Goyal, N., ... & Kiela, D. (2020). Retrieval-augmented generation for knowledge-intensive nlp tasks. (Neurlps 2020)

---

> > ### Comment · Reviewer_aWe7 · 2025-08-05
> >
> > ## Comment on A5
> > 1. RLVR typically needs answer label (outcome). Where do these labels come from in your training?
> > 2. If we have labels, there are two stages (retrieval & filter) that contribute to final reward. Is there some design like discount return? And the rollout pipeline seems very complicated since there are two models (VLM-PRF and VLM-base) interleavely involved, the paper lacks description for this pipeline.
> >
> > ## Comment on A9
> > The experimental results appear to be negative. These outcomes suggest that smaller multimodal knowledge bases correlate with higher final system accuracy, indicating that such knowledge bases may be unnecessary as they adversely impact the system rather than improve it. Alternatively, the currently proposed methodology might be insufficient to leverage the positive effects of multimodal knowledge bases.

---

> ### Author Response · Authors · 2025-08-04
>
> Dear Reviewer aWe7,
>
> Thank you again for your time and effort in reviewing our paper. As the discussion phase is drawing to a close, we would like to kindly ask whether our rebuttal has adequately addressed your concerns. If there are any remaining questions or points that need further clarification, we would be happy to provide additional details. If you feel that your concerns have been resolved, we would sincerely appreciate it if you would consider updating your score. We would also greatly value the opportunity to continue the discussion if needed.

---

> ### Author Response · Authors · 2025-08-05
>
> Dear Reviewer aWe7,
>
> Thank you for the discussion.
>
> **Q5.1:** RLVR typically needs answer label (outcome). Where do these labels come from in your training?\
> **A5.1:** The **ONLY** label is **the VQA answer**. The learning process for our Wiki-PRF, including tool-based retrieval task and information filtering task, is guided by the reward signal based solely on the correctness of the **final answer**. We consider this approach to be both fundamental and elegant. It is akin to learning to solve a maze where the only guidance comes from whether you ultimately find the exit.
>
> **Q5.2:** If we have labels, there are two stages (retrieval & filter) that contribute to final reward. Is there some design like discount return?\
> **A5.2:** **No, we don't have any discount return**. GRPO is based on importance sampling and does not require manually discounted returns. As explained in Q7, "Joint training enables the model to generate inputs explicitly optimized for downstream task performance.".
>
> **Q5.3**: The rollout pipeline seems very complicated\
> **A5.3**: We provide the following vivid example to better illustrate the rollout pipeline, where VLM-base acts purely as a tool.
>
> ```
>       +-----------+
>       |  VLM-PRF  |
>       +-----------+
>             |
>             v (Plans tool sequence)
> .------------------------------.
> |Plan: [1.Caption, 2.Grounding]|
> '------------------------------'
>             |
>             | (Execution)
>             |
>     .-------+-------.
>     |               |
>     v               v
> (1.Caption)   (2.Grounding)
> +---------------------------+
> |VLM-base (Act as the tools)|
> +---------------------------+
> ```
>
> **Q9.1**: The experimental results appear to be negative.\
> **A9.1**: Conversely, we claim that these experimental results are entirely intuitive. A larger knowledge base naturally elevates the challenge of retrieval, which in turn degrades the final VQA accuracy. This is analogous to a simple search problem: locating a single item among 100K candidates is significantly more difficult than locating it among 1K.
>
> **Q9.2**: Such knowledge bases may be unnecessary as they adversely impact the system.\
> **A9.2**:Such knowledge base is necessary. In RAG practical applications, it is common to expand a knowledge base to broaden its coverage for question answering. However, it is well-understood that a larger knowledge base is not always better; suitability is more critical. If we introduce too many irrelevant and noisy passages by overlooking the core domain of the problem, the performance of the final RAG system will inevitably be impaired. This aligns with our experimental results on the 10k, 50k, and 100k corpora.
>
> In our experiment, we focus on whether our tool-based retrieval can genuinely prove effective in slowing down the decline in QA performance as the knowledge base grows. The experimental results confirm that Wiki-PRF demonstrates significantly slower degradation for both 3B and 7B models compared to the baseline model, clearly highlighting the superiority of our method in large knowledge base settings.
>
> Thank you for reviewer's continued feedback. We look forward to discussing any further questions you may have.

---

> > ### Comment · Reviewer_aWe7 · 2025-08-06
> >
> > ### Comment on A5.3
> > Does the illustration miss the final VLM that is prompted by retrieved information and predicts the final answer?
> >
> > ### Comment on A9.2
> > To prove necessity, at least, you need to prove that there exists an interval of increasing performance within the scaling of the knowledge base from 0 to 10k.

---

> ### Author Response · Authors · 2025-08-07
>
> Dear Reviewer aWe7,
>
> Thank you for the discussion.
>
> Q5.3.1 The final VLM that is prompted by retrieved information and predicts the final answer.\
> A5.3.1 We have now supplemented the remaining part of the pipeline diagram, which we hope clarifies your concerns.
>
> ```
>                 (V,Q)
>                   |
>                   v
>             +------------+
>             |  VLM-PRF   |
>             +------------+
>                   |
>                   v
>          (Plans tool sequence)
> +------------------------------------+
> |   Plan: [1.Caption, 2.Retrieval]   |
> +------------------------------------+
>                   |
>                   | (Execution)
>                   |
>           .-------+-------.
>           |               |
>           v               v
>       (1.Caption)   (2.Grounding)
> +------------------------------------+
> |       VLM-base (Act as tools)      |
> +------------------------------------+
>                   |
>                   v
>   (Retrieves relevant passages from KB)
>             +------------+
>             |   VLM-PRF  |
>             +------------+
>                   |
>                   v
>        (Filtering information)
> +------------------------------------+
> |Information: (Filtered information) |
> +------------------------------------+
>                   |
>                   v (V,Q)
> +------------------------------------+
> | VLM-Base (Act as answer generator) |
> +------------------------------------+
>                   |
>                   v
>       +---------------------+
>       |     Final Answer    |
>       +---------------------+
> ```
>
> Q9.2.1 To prove necessity, at least, you need to prove that there exists an interval of increasing performance within the scaling of the knowledge base from 0 to 10k.\
>
> A9.2.1: Thank you again for the reviewer's feedback.
>
> *   The goal of RAG is to enhance the performance of Visual Question Answering by retrieving and recalling the passages from **a given knowledge base** \[1\]\[2\]\[3\]. However, the discussion seems to focuse on the necessity of RAG, ie. **the scale of knowledge base**.  Our summary on this issue is as follows: In the Q9, we ensured that the correct document corresponding to each evaluation question was included in knowledge bases of all scales.Therefore, increasing the scale of knowledge base raises retrieval difficulty and weaken the effectiveness of RAG. In addition, We add an experiment with 10k (random) in Table R3. The results show that when the correct document is not guaranteed to be included, increasing the scale of knowledge base helps retrieve relevant documents, thereby enhancing RAG performance.
>
>
> **Table R3:** Ablation studies on the scale of knowledge base on InfoSeek.
>
> | Method | Model | 0 | 10k (random) | 10K(GT included) |
> | --- | --- | --- | --- | --- |
> | Vanilla-MRAG | Qwen2.5-VL-3B | 19.3% | 24.2% | 49.7% |
> | Wiki-PRF-3B | VLM-PRF-3B | 23.9% | 29.6% | 53.0% |
> | Vanilla-MRAG | Qwen2.5-VL-7B | 20.1% | 27.3% | 56.3% |
> | Wiki-PRF-7B | VLM-PRF-7B | 28.3% | 36.1% | 60.3% |
>
> *   Since RAG methods are typically evaluated by comparing the accuracy of VQA responses under a given knowledge base, in Q9 we compare the performance of our method against Vanilla-MRAG under knowledge bases (contain the correct document.) of different sizes.Table R2 shows that our method consistently outperforms Vanilla-MRAG across knowledge bases (contain the correct document) of different sizes. Moreover, we observe that, with larger knowledge bases (50k, 100k), our method exhibits a smaller performance drop compared to Vanilla-MRAG, indicating its superior ability to retrieve relevant document from knowledge base.
>
>
> *   Finally, supported by both the original experiments in the paper and the additional results in the rebuttal, we have conducted comprehensive generalization evaluations across diverse datasets (knowledge base sizes of 115K (OK-VQA), 100K (InfoSeek), and 2M (E-VQA)) and under different model scales, including Qwen2.5-VL (3B and 7B) and InternVL3 (8B). Our method consistently achieves SOTA performance. These results strongly demonstrate the superiority and strong generalizability of our method.
>
>
> \[1\] Ling, Zihan, et al. "MMKB-RAG: A Multi-Modal Knowledge-Based Retrieval-Augmented Generation Framework." (arXiv 2025).
>
> \[2\] Yan, Yibin, and Weidi Xie. "EchoSight: Advancing visual-language models with Wiki knowledge." (EMNLP 2024).
>
> \[3\] Cocchi, Federico, et al. "Augmenting multimodal llms with self-reflective tokens for knowledge-based visual question answering." (_CVPR_ 2025.)

---

> ### Author Response · Authors · 2025-08-09
> **Acknowledgments**
>
> Dear Reviewer aWe7,
>
> As our discussion nears its conclusion, we would like to sincerely thank you for your thoughtful comments and discussion. If your concerns have been addressed, we would be truly grateful if you could kindly consider raising the score. If you have any further questions, we are pleased to address your concerns.

---

### Note · Authors · 2025-08-13

Dear Reviewers and Area Chair,

We sincerely thank all reviewers and the Area Chair for the invaluable feedback and efforts throughout the review and rebutal process. We are encouraged by the reviewers' positive feedback on our Wiki-PRF, particularly their recognition of our clear motivation (Reviewer CtMV), novelty (Reviewers bmZr, VEoU), well-written presentation (Reviewer CtMV), and strong/comprehensive experimental results (Reviewers aWe7, bmZr, VEoU).

To address the reviewers' concerns, we have conducted  extensive experiments and detailed analyses. Below, we summarize key responses to concerns raised during the rebuttal process:

● Scalability (Reviewer aWe7): Experiments on varying scales (10k, 50k, and 100k in Tab. R2 and R3) demonstrate that our method consistently achieves strong performance across knowledge bases of different scales.

● Generalizability (Reviewer bmZr): The strong performance of Wiki-PRF on OK-VQA benchmark (77.8% in Tab R2) demonstrate the effectiveness of our work. Moreover, additional experiments show that our method deliver consistent results across diverse base models (Tab. R1) .

● Effectiveness of RL (Reviewer VEoU): Compared to SFT-based filtering, the RL model better captures the information during filtering, resulting in significantly improved robustness and generalizability (46.3% vs. 41.8% in Tab. R1).

● Evaluation Protocol (Reviewer CtMV): We evaluate our method via the VQA score on the OK-VQA benchmark. Evaluation details are provided to enable replication.

We hope that these experiments and explanations can clarify the capabilities and effectiveness of Wiki-PRF. We will fully open-source our implementation to enable further validation and reproducibility.

Yours sincerely,\
Authors

---

### Decision · Program_Chairs · 2025-09-17

**Decision:**

Accept (poster)

**Comment:**

Existing knowledge base visual question answering (KB-VQA) systems suffer from insufficient interactivity during knowledge retrieval and ineffective organization of retrieved information for Visual-Language Model (VLM). To address these issues, this paper proposes a three-stage visual language model based on the Process, Retrieve, Filter (VLM-PRF) framework.


This paper proposes Wiki-PRF, a three-stage (processing–search–filtering) pipeline that is highly effective at addressing the challenges of multimodal RAG in KB-VQA. The proposed method combines reinforcement learning and tool utilization. It is the first attempt in this field and provides new insights into query expansion and knowledge representation selection. Comprehensive experiments on E-VQA and InfoSeek, along with ablation studies, demonstrate the effectiveness of each stage and tool by consistently outperforming strong baselines such as ReflectiVA and MMKB-RAG.


The author's detailed response and new experiments have addressed many of the concerns that were raised. The effectiveness of the proposed method and its conclusions (that MM RAG can improve RLVR and is useful for knowledge-based VQA) are considered convincing. As a result, multiple reviewers have maintained a positive stance, deeming the paper worthy of acceptance. However, one reviewer issued a borderline rejection. This reviewer's main concern pertains to the evaluation setup of the VQA dataset. The AC has thoroughly reviewed this discussion and believes that the authors have appropriately addressed the reviewer's concerns.


Based on the author's careful rebuttal and additional experiments, the concerns have been resolved, and the proposed method and conclusions are considered convincing. Therefore, the AC has decided to accept the paper, and it is strongly recommended that the reviewers' comments be reflected in the camera-ready version.